# Learning Patient-Specific Disease Dynamics with Latent Flow Matching for Longitudinal Imaging Generation

**Hao Chen**[1*]   **Rui Yin**[2*]   **Yifan Chen**[1]   **Qi Chen**[3]   **Chao Li**[1,4]
[1]University of Cambridge,   [2]Nanjing First Hospital, Nanjing Medical University
[3]Johns Hopkins University   [4]University of Dundee

## Abstract

Understanding disease progression is a central clinical challenge with direct implications for early diagnosis and personalized treatment. While recent generative approaches have attempted to model progression, key mismatches remain: disease dynamics are inherently continuous and monotonic, yet latent representations are often scattered, lacking semantic structure, and diffusion-based models disrupt continuity with a random denoising process. In this work, we propose to treat the disease dynamics as a velocity field and leverage Flow Matching (FM) to align the temporal evolution of patient data. Unlike prior methods, it captures the intrinsic dynamics of disease, making the progression more interpretable. However, a key challenge remains: in latent space, autoencoders (AEs) do not guarantee alignment across patients or correlation with clinical-severity indicators (*e.g.*, age and disease conditions). To address this, we propose to learn patient-specific latent alignment, which enforces patient trajectories to lie along a specific axis, with magnitude increasing monotonically with disease severity. This leads to a consistent and semantically meaningful latent space. Together, we present $\Delta$-LFM, a framework for modeling patient-specific latent progression with flow matching. Across three longitudinal MRI benchmarks, $\Delta$-LFM demonstrates strong empirical performance and, more importantly, offers a new framework for interpreting and visualizing disease dynamics.

## 1 Introduction

Modeling disease progression is a fundamental problem in healthcare (Suk & Shen, 2013; Wang et al., 2020), with profound implications for early detection and individualized treatment. Disease does not evolve uniformly: differences in anatomy, genetics, and environment give rise to heterogeneous trajectories, which are an intrinsic property of disease dynamics (Real & Biek, 2007; Parratt et al., 2016; Li et al., 2025). Accounting for this heterogeneity is essential for accurately characterizing temporal changes, identifying preclinical biomarkers, and designing personalized therapies that can improve clinical outcomes (Frisoni et al., 2010). However, most progression models capture only population-level trends, overlooking individualized variation (Rokuss et al., 2025; Yang et al., 2025b). This limitation obscures early-stage signals and misaligns disease severity with underlying anatomy, ultimately reducing clinical utility. Capturing patient-specific dynamics is therefore critical, but remains an underexplored challenge (Lai et al., 2025).

Recent advances in generative modeling have opened new opportunities for studying disease progression. Traditional biomarker-based approaches often reduce complexity to coarse measures (*e.g.*, volume changes (Jack Jr et al., 2003; Tabrizi et al., 2011) or disease rating (O'Bryant et al., 2008; Young et al., 2020)), which can obscure heterogeneous disease dynamics. In contrast, longitudinal medical image generation offers a principled framework for visualizing symptomatic transitions and individualized patterns of anatomical change. Rather than limiting analysis to numeric outcomes, synthesized images yield interpretable and more informative visual representations that

---

*Equal contribution. Correspondence to: hc666@cam.ac.uk.

support structured analyses and offer more actionable insights for clinical practice (Wu et al., 2024; Chen et al., 2024; 2025).

Several methods have attempted to model disease progression. For instance, DiffuseMorph (Kim et al., 2022) introduces a morph field to describe pixel-wise displacements, and TADM (Litrico et al., 2024) incorporates temporal awareness by predicting residual images conditioned on age gaps. Despite these advances, generating temporally consistent and individualized disease trajectories remains an open problem. Beyond high-fidelity image synthesis, it is crucial for generative models to preserve fine-grained anatomical structures and accurately capture patient-specific dynamics. Although BrLP (Puglisi et al., 2024) takes a step toward personalization by modeling volume changes, such guidance is coarse and provides only limited control over individualized trajectories.

**Contributions:** Motivated by these gaps, we propose a personalized generative framework for modeling disease progression based on flow matching. Our approach, termed Progression Latent Flow Matching ($\Delta$-LFM), is designed to generate trajectories at arbitrary future time points within a patient-specific latent space, ensuring temporal consistency and clinical interpretability. The key contributions are summarized as follows:

- We reformulate conventional flow matching, which typically models dynamics from $t = 0$ to $t = 1$, into a formulation where the time variable $t$ explicitly encodes the future time gap. This extension transforms the horizon from a normalized range $[0, 1]$ to a meaningful temporal range $[0, T]$, enabling prediction at arbitrary future time points $T$ with consistent temporal semantics.

- We introduce a patient-specific latent space, regularized by a new *ArcRank* loss, which enforces chronology-aware alignment and captures individualized disease dynamics.

- We show that the learned latent space supports interpretable visualization of patient trajectories. Notably, although disease severity is not used for training, the latent representations naturally reflect severity levels, providing clinically meaningful structure.

- We further validate $\Delta$-LFM on longitudinal MRI benchmarks and propose a progression-specific evaluation metric tailored to disease modeling called $\Delta$-RMAE. Results demonstrate improved imaging fidelity and more accurate alignment with actual disease progression.

## 2 RELATED WORK

**Generative models for disease progression.** Generative modeling has emerged as a powerful strategy for simulating longitudinal disease changes. Early efforts relied on generative adversarial networks (GANs), either by introducing morphological priors to simulate aging effects (Ravi et al., 2019) or by predicting deformation fields with 3D conditional GANs instead of direct intensity values (Ravi et al., 2022).

More recently, diffusion models (Ho et al., 2020) have become the dominant paradigm due to their superior fidelity and stability. Extensions of this framework introduce temporal awareness and anatomical constraints. Sequence-Aware Diffusion Model (SADM) leverages multiple scans to autoregressively generate future MRIs (Yoon et al., 2023), while DiffuseMorph (Kim et al., 2022) produces smooth voxel-wise deformation fields across timepoints, thereby preserving topology. Progressive Image Editing (PIE) (Liang et al., 2023) edits disease-related regions in medical images using text-guided diffusion models to reflect how a patient's condition evolves over time. Kyung *et al.* (Kyung et al., 2024) adopt electronic health records (EHRs), including medical history, to model temporal changes and enable medically meaningful generation. Temporally-Aware Diffusion Model (TADM) conditions on age maps and adopts a residual prediction strategy, modeling progression as incremental changes rather than absolute reconstructions (Litrico et al., 2024).

Conditioning strategies further enhance subject specificity and anatomical fidelity. Brain Latent Progression (BrLP) employs ControlNet with volumetric ratio conditioning to synthesize individualized disease trajectories while integrating population priors (Puglisi et al., 2024; 2025). Similarly, BrainMRDiff fuses multiple structural masks (*e.g.*, white matter, gray matter, ventricles) into a unified conditioning representation, substantially improving anatomical realism (Bhattacharya et al., 2025). ImageFlowNet (Liu et al., 2025a) learns multiscale joint patient embeddings and position-parameterized neural flow fields to forecast full image-level trajectories.

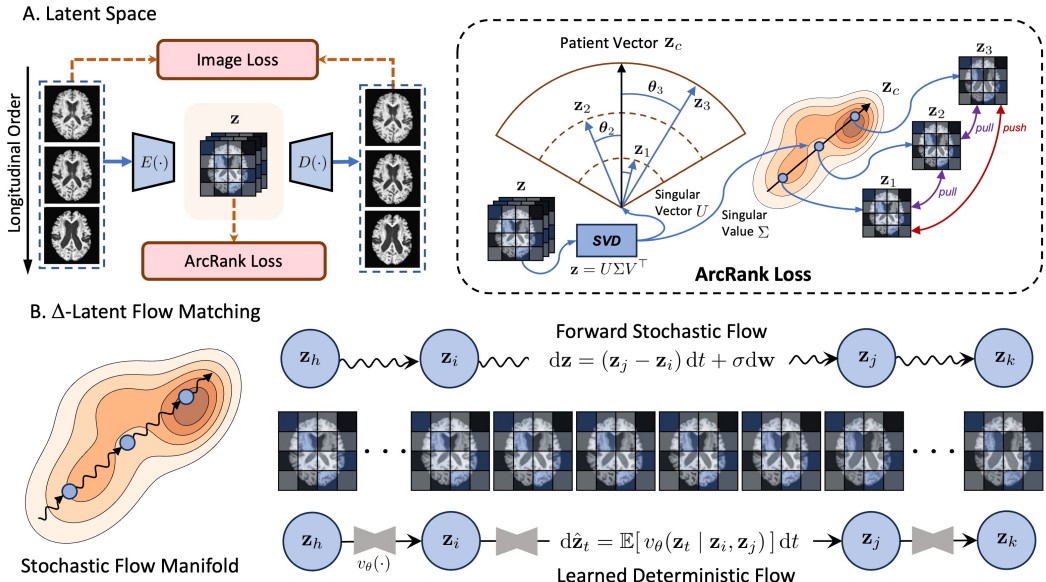

Figure 1: $\Delta$-LFM overview. $\Delta$-LFM operates in the latent space. In the first stage, an autoencoder constructs the latent space with ArcRank Loss to capture patient-specific disease trajectories. In the second stage, flow matching predicts disease progression along the trajectory over the total time horizon $T/\mathrm{d}t$, where, for example, $T = j - i$, to adapt the actual prediction span rather than relying on a fixed interval.

In contrast to these approaches, which primarily focus on realism and population-informed priors, our method targets patient-specific disease trajectories while explicitly modeling heterogeneity through a flow-matching generative framework.

**Subject-level consistency and biomedical plausibility.** Ensuring subject-specific consistency and biomedical plausibility is a central challenge in modeling synthetic disease progression. Early approaches such as the Longitudinal Variational Autoencoder (Ramchandran et al., 2021) and Longitudinal Self-Supervised Learning (LNE) (Zhao et al., 2021) focus on learning patient-temporal dynamics by constructing temporally coherent latent spaces, thereby producing individualized representations for downstream tasks.

Subsequent works introduced explicit mechanisms to regularize longitudinal consistency. Ren *et al.* (Ren et al., 2023) enforce subject-level coherence through local and multi-scale spatio-temporal features within a self-supervised framework. In a complementary direction, Treatment-Aware Diffusion (TaDiff) (Liu et al., 2025b) incorporates prior scans and treatment information to jointly generate future MRIs and tumor segmentations, thereby improving biological plausibility by conditioning on clinically relevant variables.

## 3 METHODOLOGY

Our proposed approach consists of two key components: (i) patient-specific latent learning via *ArcRank* (Sec. 3.2), and (ii) arbitrary-time progression modeling via $\Delta$-LFM (Sec. 3.3).

Before delving into these components, we first present the preliminaries in Sec. 3.1.

### 3.1 PRELIMINARIES

Flow matching (Lipman et al., 2023; Tong et al., 2024) trains continuous normalizing flows by directly aligning vector fields instead of relying on stochastic denoising objectives as in diffusion models. Let $\pi_0$ and $\pi_1$ denote the source and target distributions, respectively. Flow matching defines a family of intermediate distributions $\{\pi_t\}_{t \in [0,1]}$ along a continuous trajectory connecting

$\pi_0$ and $\pi_1$. The goal is to learn a time-dependent velocity field $v_\theta(\mathbf{x}, t)$ such that the solution of the ordinary differential equation (ODE) (Lu et al., 2022) transports $\pi_0$ exactly to $\pi_1$ at $t = 1$:

$$\frac{d\mathbf{x}_t}{dt} = v_\theta(\mathbf{x}_t, t), \tag{1}$$

where the velocity field $v_\theta(\mathbf{x}, t)$ is trained to ensure that samples from $\pi_0$ flow continuously into $\pi_1$.

Unlike score-based diffusion models, which approximate the score function $\nabla_\mathbf{x} \log \pi_t(\mathbf{x})$ via noisy denoising objectives, flow matching bypasses score estimation by directly regressing the neural vector field $v_\theta(\mathbf{x}, t)$ onto analytically known target fields. For example, under conditional flow matching (Lipman et al., 2023), one defines a reference path $\mathbf{x}_t = (1 - t)\mathbf{x}_0 + t\mathbf{x}_1$ between samples $\mathbf{x}_0 \sim \pi_0$ and $\mathbf{x}_1 \sim \pi_1$, yielding a target velocity

$$v^\star(\mathbf{x}_t, t) = \mathbf{x}_1 - \mathbf{x}_0, \tag{2}$$

which is constant along the trajectory. The training objective then minimizes

$$\mathcal{L}_{\text{FM}}(\theta) = \mathbb{E}_{\mathbf{x}_0 \sim \pi_0, \mathbf{x}_1 \sim \pi_1, t \sim \mathcal{U}[0,1]} \big\| v_\theta(\mathbf{x}_t, t) - v^\star(\mathbf{x}_t, t) \big\|^2, \tag{3}$$

which yields stable training and ensures the learned flow recovers the correct marginal $\pi_1$ at $t = 1$.

To further incorporate stochasticity (Albergo et al., 2023), the deterministic flow can be extended to a stochastic differential equation of the form

$$d\mathbf{z}_t = v^\star(\mathbf{z}_t, t)\,dt + \sigma\,d\mathbf{w}_t, \tag{4}$$

where $\mathbf{w}_t$ is a Wiener process and $\sigma$ controls the noise scale. This extension allows flow matching to model both deterministic transport and stochastic variability, providing greater flexibility in capturing complex data distributions.

## 3.2 ARCRANK: LONGITUDINAL ALIGNMENT OF PATIENT LATENT

To enhance sensitivity to patient-specific dynamics, we introduce a chronologically-aware latent representation learned via a new contrastive objective, termed *ArcRank Loss*. This objective aligns representations across time for the same individual by enforcing both angular consistency and temporal ordering in the embedding space. Let $\mathbf{z}$ denote the latent representation of an input $\mathbf{x}$, extracted from a Variational Autoencoder (VAE) (Kingma & Welling, 2013) encoder $E(\cdot)$. The loss encourages

$$\angle(\mathbf{z}_i) \approx \angle(\mathbf{z}_j) \quad \wedge \quad \|\mathbf{z}_i\| \prec \|\mathbf{z}_j\| \quad \text{if } t_i < t_j, \qquad \mathbf{z}_i, \mathbf{z}_j \in \mathcal{P}^k, \tag{5}$$

where $\mathcal{P}^k$ denotes the latent feature set of patient $k$, and $t$ here represents the imaging capture time. Here, $\angle(\cdot)$ represents the direction of a latent vector, and $\|\cdot\|$ is its magnitude. Intuitively, samples from the same patient are encouraged to align along a consistent latent direction, while the trajectory respects chronological order (further along the trajectory corresponds to later disease stages).

In practice, we extract these components via Singular Value Decomposition (SVD) (Klema & Laub, 1980) and write

$$\angle\mathbf{z} = U, \quad \|\mathbf{z}\| = \Sigma, \quad \text{where } U\Sigma V^\top = \text{SVD}(\mathbf{z}). \tag{6}$$

where $U$ captures the orientation (angle) and $\Sigma$ encodes the scaling (norm).

**ArcRank Loss Computation.** For a given patient $k$, let $\mathcal{P}^k = \{\mathbf{z}_1, \mathbf{z}_2, \cdots, \mathbf{z}_n\}$ denote latent features at times $t_1 < t_2 < \cdots < t_n$. We decompose each feature via SVD:

$$\angle(\mathbf{z}_t) = U_t, \qquad \|\mathbf{z}_t\| = \Sigma_t. \tag{7}$$

The angular consistency loss encourages stability of feature orientations across time, while the temporal ranking loss enforces monotonic growth of feature magnitudes:

$$\mathcal{L}_{\text{Arc}} = \sum_{\substack{i<j \\ \text{same patient}}} |U_i - U_j|, \quad \mathcal{L}_{\text{Rank}} = \sum_{\substack{i<j \\ \text{same patient}}} \max\big(0, m - (\Sigma_j - \Sigma_i)\big), \quad t_i < t_j, \tag{8}$$

where $m > 0$ is a margin hyperparameter. The final objective combines the two components:

$$\mathcal{L}_{\text{ArcRank}} = \lambda_{\text{arc}}\,\mathcal{L}_{\text{Arc}} + \lambda_{\text{rank}}\,\mathcal{L}_{\text{Rank}}, \tag{9}$$

where $\lambda_{\text{arc}}, \lambda_{\text{rank}} > 0$ balance the contributions.

In practice, the ranking term $\mathcal{L}_{\text{Rank}}$ tends to push timepoints for the same patient increasingly far apart, and the margin $m$ is shared across all pairs $(i, j)$, i.e., it is not adaptively optimized for different temporal gaps. To mitigate this effect, we introduce an additional pull term that penalizes excessive separation in magnitude between temporally adjacent features:

$$\mathcal{L}_{\text{Pull}} = \left| \Sigma_j - \Sigma_i \right|. \tag{10}$$

We augment the ranking objective as

$$\tilde{\mathcal{L}}_{\text{Rank}} = \mathcal{L}_{\text{Rank}} + \mathcal{L}_{\text{Pull}}, \tag{11}$$

using the same overall weight as $\mathcal{L}_{\text{Rank}}$ in the total loss. This pull term softly encourages consecutive timepoints to remain close in latent space, while still allowing the hinge-based push term to enforce the desired temporal ordering.

To stabilize training, for each pair $(i, j)$ used in the ArcRank loss, we apply the stop-gradient operator $\text{sg}(\cdot)$ to the representation of $i$.

### 3.3 $\Delta$-LFM: Flow Matching along Patient Trajectories

While ArcRank enforces patient-specific chronological alignment, it does not explicitly capture how latent states evolve over time. To address this limitation, we introduce $\Delta$-LFM, a flow-matching objective that learns smooth mappings across arbitrary time intervals within a patient's trajectory. Unlike standard flow matching, which interpolates between fixed source and target distributions (*e.g.*, from $0 \to 1$), $\Delta$-LFM models temporal transitions on disease-specific time scales $T$, where $T$ denotes the future time relative to the baseline. For clarity, we refer to the conventional $[0, 1]$ interval as *Physical Sampling*, and to our proposed $[0, T]$ formulation as *Temporal Sampling*.

Formally, given a patient $k$, let $< \mathbf{z}_i, \mathbf{z}_j >$ denote paired latent states at times $t_i < t_j$. We define the latent transition field $v_\theta(\cdot)$, parameterized by a neural network, which approximates the continuous-time velocity of latent dynamics. The goal is to match this learned velocity field to the empirical difference between latent features across time:

$$v^\star(i, j) \approx \frac{\mathbf{z}_j - \mathbf{z}_i}{t_j - t_i}, \quad t_j > t_i, \tag{12}$$

The associated flow-matching loss is then given by

$$\mathcal{L}_{\text{LFM}}(\theta) = \sum_{i<j} \left| v_\theta(i, j) - v^\star(i, j) \right|^2. \tag{13}$$

**Inference.** Given an input state $\mathbf{x}_i$ observed at time $t_i$, the encoder maps it to a latent representation $\mathbf{z}_i$. To predict the target state $\mathbf{x}_j$ at a later time $t_j$, $\Delta$-LFM integrates the learned velocity field forward in time. Specifically, we fix a discretization step size $\text{d}t > 0$ (*e.g.*, $\text{d}t = 0.01$), so that the total number of integration steps is

$$N = \frac{t_j - t_i}{\text{d}t}. \tag{14}$$

At each step, the latent state is updated according to the learned dynamics

$$\mathbf{z}_{i+\text{d}t} = \mathbf{z}_i + \text{d}t \cdot v_\theta(\mathbf{z}_i, t_i), \tag{15}$$

with time advanced as $t_{i+\text{d}t} = t_i + \text{d}t$. Repeated application of this update yields an approximation of the latent state at $t_j$, which is then decoded to produce the predicted observation $\hat{\mathbf{x}}_j$.

### 3.4 Discussion

**The use of Flow Matching.** We use flow matching because it naturally models continuous trajectories while keeping intermediate states interpretable, which is crucial in our setting. It learns a continuous velocity field between observed endpoints and allows direct sampling of intermediate states, without relying on long numerical integrations. Compared to latent ODEs, this avoids sensitivity to ODE solver choices and reduces computational cost. While rectified flow methods are effective, they are typically built on diffusion trajectories; in contrast, flow matching gives a more direct and tractable formulation of the probability path that is better aligned with our goal of faithful temporal interpolation.

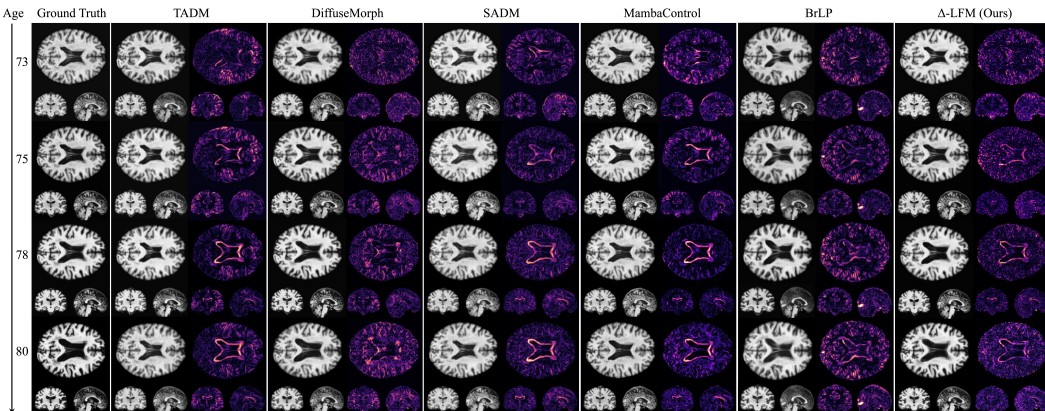

Figure 2: **Method comparison.** The first column shows the ground truth. All predictions are initialized from the same baseline scan at age 71. Odd columns show predicted MRIs; even columns show residual maps relative to the ground truth. Residuals use the `magma` colormap, scaled to the 1st to 99th percentiles of the residual distribution, enhancing error visibility around the lateral ventricles (central butterfly-shaped region) and cortical gray matter (outer surface).

**Latent-space trajectories reasoning.** One central assumption in our framework is that disease progression can be represented as a straight line in latent space, structured along a linear trajectory where the magnitude encodes disease stage and the direction preserves patient identity. While this design simplifies prediction, it raises an important question: can a straight line adequately capture both the shared and divergent aspects of disease dynamics across patients? We argue that this concern is valid. In our work, however, the linear trajectory is not intended to reproduce the full biological complexity of disease, but rather to provide a stable and interpretable scaffold that highlights the primary mode of variation corresponding to disease advancement.

**Temporal uneven progression.** We formulate flow matching (FM) over the interval $[0, T]$, interpreting the index as a progression coordinate rather than physical clock time. However, although this formulation accounts for time differences, it implicitly assumes that progression is evenly distributed along the disease trajectory. In practice, progression is often uneven: extended periods of stability may be punctuated by sudden changes, or vice versa. The constant-velocity assumption in FM fails to capture this heterogeneity, effectively treating patient scans as uniformly spaced along the trajectory. To address this limitation, we introduce conditioning on the target time $T$ together with patient-specific attributes (*e.g.*, sex, age, and clinical status) to enable the model to capture heterogeneous progression patterns that vary substantially over time.

**Design details of ArcRank loss.** The proposed ArcRank loss uses SVD to jointly capture the direction and magnitude of latent trajectories. Alternatives such as cosine similarity for angles and absolute values for magnitudes treat these aspects separately and can become unstable when latent features are noisy or vary in scale. In contrast, SVD provides an orthogonal basis in which singular vectors naturally describe trajectory directions and singular values represent their magnitudes. This unified representation is compact and numerically stable, allowing ArcRank loss to align trajectories along meaningful progression directions while penalizing inconsistent scaling.

## 4 EXPERIMENTS

### 4.1 EXPERIMENT SETTINGS

We build our study on three major Alzheimer's disease (AD) cohorts: ADNI (Petersen et al., 2010), OASIS-3 (LaMontagne et al., 2019), and AIBL (Ellis et al., 2009). Every MRI undergoes the same preprocessing pipeline: N4 bias correction (Tustison et al., 2010), skull stripping (Hoopes et al., 2022), resampling to 1.5 mm³ voxels. We then normalize tissue contrasts and register all scans to a refined standard brain template. We use a random 80/5/15 split for train/validation/test.

Table 1: Quantitative comparison of methods in terms of image quality metrics, including PSNR (dB) and **SSIM** ($\times 1.0E2$). Results are reported as mean $\pm$ standard deviation. *Type* denotes the prediction strategy: "Direct" generates outputs directly, "Deform" estimates a transformation field, "Noise" applies diffusion-based denoising, "$\Delta$Noise" predicts and denoises progression residuals, and "$\Delta$Flow" models progression through gradual residual prediction.

| Method | Type | ADNI | | AIBL | | OASIS | |
|---|---|---|---|---|---|---|---|
| | | PSNR $\uparrow$ | SSIM $\uparrow$ | PSNR $\uparrow$ | SSIM $\uparrow$ | PSNR $\uparrow$ | SSIM $\uparrow$ |
| CardiacAging (Campello et al., 2022) | Direct | $27.78 \pm 1.49$ | $92.04 \pm 0.99$ | $28.41 \pm 1.41$ | $90.30 \pm 0.90$ | $26.23 \pm 1.45$ | $86.17 \pm 2.06$ |
| CounterSynth (Pombo et al., 2023) | Direct | $28.24 \pm 1.31$ | $92.84 \pm 0.95$ | $28.96 \pm 1.25$ | $92.10 \pm 0.91$ | $26.79 \pm 1.42$ | $86.45 \pm 1.98$ |
| DiffuseMorph (Kim et al., 2022) | Deform | $29.56 \pm 1.63$ | $93.57 \pm 0.93$ | $29.17 \pm 1.54$ | $92.70 \pm 1.02$ | $28.13 \pm 1.36$ | $88.16 \pm 1.69$ |
| SADM (Yoon et al., 2023) | Noise | $26.94 \pm 2.28$ | $85.15 \pm 2.72$ | $27.97 \pm 1.91$ | $89.40 \pm 1.78$ | $26.74 \pm 1.80$ | $86.36 \pm 2.37$ |
| TADM (Litrico et al., 2024) | $\Delta$Noise | $27.89 \pm 1.91$ | $90.78 \pm 1.54$ | $28.58 \pm 1.77$ | $91.70 \pm 1.01$ | $26.29 \pm 1.52$ | $85.51 \pm 1.93$ |
| ImageFlowNet (Liu et al., 2025a) | Flow | $28.37 \pm 1.18$ | $92.96 \pm 0.88$ | $29.08 \pm 1.12$ | $92.23 \pm 0.84$ | $27.92 \pm 1.29$ | $87.63 \pm 1.82$ |
| BrLP (Puglisi et al., 2024) | Noise | $28.51 \pm 1.77$ | $91.52 \pm 1.31$ | $28.96 \pm 1.36$ | $92.21 \pm 0.97$ | $27.98 \pm 1.40$ | $87.93 \pm 1.44$ |
| MambaControl (Yang et al., 2025a) | Noise | $29.72 \pm 1.04$ | $93.60 \pm 0.96$ | $29.86 \pm 1.21$ | $93.17 \pm 0.98$ | $28.24 \pm 1.28$ | $88.30 \pm 1.52$ |
| $\Delta$-LFM (Ours) | $\Delta$Flow | $\mathbf{30.59 \pm 0.89}$ | $\mathbf{94.62 \pm 0.85}$ | $\mathbf{30.52 \pm 1.03}$ | $\mathbf{93.92 \pm 0.88}$ | $\mathbf{29.01 \pm 1.19}$ | $\mathbf{89.36 \pm 1.29}$ |

We benchmark against a broad spectrum of recent approaches. Direct-prediction methods include CardiacAging (Campello et al., 2022) and CounterSynth (Pombo et al., 2023). Deformation-based modeling represented by DiffuseMorph (Kim et al., 2022). Diffusion-based methods span multiple variants: SADM (Yoon et al., 2023), which denoises full images; TADM (Litrico et al., 2024), which denoises residuals; and two ControlNet-based (Zhang et al., 2023) models, BrLP (Puglisi et al., 2024) and MambaControl (Yang et al., 2025a).

**Metrics.** We evaluate reconstruction quality with two standard image-level metrics: Peak Signal-to-Noise Ratio (PSNR) and Structural Similarity Index Measure (SSIM). To capture anatomical fidelity, we followed BrLP (Puglisi et al., 2024) to report region-level mean absolute error (MAE) over clinically relevant structures (hippocampus, amygdala, lateral ventricles, cerebrospinal fluid (CSF), and thalamus).

However, measuring image similarity alone is insufficient in the longitudinal setting. Scans from the same patient naturally exhibit high similarity because they share the same biological structure. The subtle deformations caused by disease are often small and easily overshadowed by the stable, unaffected anatomy. As a result, conventional metrics such as PSNR and SSIM tend to report inflated scores, failing to reflect clinically meaningful progression.

The true signal of interest lies in the *residual differences* between baseline and follow-up scans. These residuals often encode the disease trajectory. To capture this, we introduce a residual-based metric. Specifically, let

$$\Delta_{\text{gt}} = \mathbf{x}_T - \mathbf{x}_0, \quad \Delta_{\text{gen}} = \hat{\mathbf{x}}_T - \mathbf{x}_0, \tag{16}$$

where $\Delta$ denotes the residual image, $\mathbf{x}$ and $\hat{\mathbf{x}}$ denote ground-truth and generated MRIs, respectively. The metric is defined as the Residual-based Relative Mean Absolute Error ($\Delta$-RMAE):

$$\Delta\text{-RMAE} = \frac{|\Delta_{\text{gt}} - \Delta_{\text{gen}}|}{\frac{1}{2}\left(|\Delta_{\text{gt}}| + |\Delta_{\text{gen}}|\right)}. \tag{17}$$

By definition, $\Delta$-RMAE $\in [0, 2]$. A smaller value indicates that the predicted residual aligns with the ground-truth residual, *i.e.*, the model has correctly captured disease-relevant evolution. If the model predicts almost no change (essentially copying the baseline image), then $\Delta_{\text{gen}} \approx 0$. In this case, both numerator and denominator are dominated by $|\Delta_{\text{gt}}|$, and the score approaches 2. Likewise, if the prediction deviates strongly, $|\Delta_{\text{gen}}|$ dominates, again driving the score toward 2.

## 4.2 BENCHMARK COMPARISONS

We present qualitative results in Figure 2. In this subject, disease progression primarily affects the lateral ventricles (the central butterfly-shaped region) and the cortical gray matter (outer surface). SADM, TADM, and MambaControl fail to capture changes in the lateral ventricles, resulting in pronounced butterfly-shaped error in the residual image. DiffuseMorph and BrLP better capture ventricular changes, but leave substantial residuals along the cortical surface. Our method also

Table 2: Quantitative comparison results on clinical structure faithfulness metrics are reported in region-based MAE and $\Delta$-RMAE (Residual-based Relative Mean Absolute Error). Results are reported as mean $\pm$ standard deviation.

| Method | Type | ADNI | | AIBL | | OASIS | |
|---|---|---|---|---|---|---|---|
| | | Region MAE ↓ | Δ-RMAE ↓ | Region MAE ↓ | Δ-RMAE ↓ | Region MAE ↓ | Δ-RMAE ↓ |
| CardiacAging (Campello et al., 2022) | Direct | 0.289 ± 0.33 | 0.771 ± 0.12 | 0.281 ± 0.35 | 0.698 ± 0.13 | 0.345 ± 0.35 | 0.739 ± 0.13 |
| CounterSynth (Pombo et al., 2023) | Direct | 0.272 ± 0.32 | 0.704 ± 0.11 | 0.256 ± 0.30 | 0.672 ± 0.12 | 0.298 ± 0.34 | 0.690 ± 0.12 |
| DiffuseMorph (Kim et al., 2022) | Deform | 0.230 ± 0.28 | 0.516 ± 0.10 | **0.226 ± 0.27** | 0.482 ± 0.10 | 0.279 ± 0.29 | 0.503 ± 0.11 |
| SADM (Yoon et al., 2023) | Noise | 0.283 ± 0.34 | 0.746 ± 0.11 | 0.278 ± 0.30 | 0.693 ± 0.12 | 0.365 ± 0.39 | 0.728 ± 0.12 |
| TADM (Litrico et al., 2024) | ΔNoise | 0.263 ± 0.32 | 0.697 ± 0.11 | 0.243 ± 0.30 | 0.654 ± 0.11 | 0.291 ± 0.34 | 0.685 ± 0.11 |
| ImageFlowNet (Liu et al., 2025a) | Flow | 0.259 ± 0.30 | 0.589 ± 0.11 | 0.240 ± 0.29 | 0.561 ± 0.12 | 0.283 ± 0.32 | 0.574 ± 0.12 |
| BrLP (Puglisi et al., 2024) | Noise | 0.257 ± 0.32 | 0.630 ± 0.10 | 0.278 ± 0.31 | 0.594 ± 0.11 | 0.335 ± 0.36 | 0.622 ± 0.11 |
| MambaControl (Yang et al., 2025a) | Noise | 0.225 ± 0.30 | 0.554 ± 0.09 | 0.249 ± 0.30 | 0.525 ± 0.10 | 0.299 ± 0.32 | 0.561 ± 0.10 |
| Δ-LFM (Ours) | ΔFlow | **0.210 ± 0.28** | **0.436 ± 0.08** | 0.226 ± 0.29 | **0.417 ± 0.10** | **0.262 ± 0.30** | **0.473 ± 0.08** |

Table 3: Ablation study results (mean across datasets). We evaluate the effect of different auto-encoder training strategies (Arc Loss, Rank Loss, ArcRank Loss) and flow-matching manifold sampling strategies ($[0, 1]$ vs. $[0, T]$). The Cond. represent using condition or not. The row with gray background indicated the proposed method.

| Model | Auto-Encoder | FM Sampling | Cond. | Image Quality | | Structure Fidelity | |
|---|---|---|---|---|---|---|---|
| | | | | PSNR ↑ | SSIM ↑ | Region MAE ↓ | Δ-RMAE ↓ |
| LFM (Baseline) | - | [0, 1] | ✗ | 27.59 ± 1.78 | 88.62 ± 1.63 | 0.357 ± 0.342 | 0.552 ± 0.146 |
| Conditional LFM | - | [0, 1] | ✓ | 28.46 ± 1.22 | 91.20 ± 1.15 | 0.299 ± 0.312 | 0.486 ± 0.101 |
| | - | [0, T] | ✓ | 28.78 ± 1.15 | 90.97 ± 1.07 | 0.286 ± 0.295 | 0.472 ± 0.091 |
| | w/ Arc Loss | [0, 1] | ✓ | 29.52 ± 0.98 | 92.38 ± 0.95 | 0.251 ± 0.282 | 0.457 ± 0.086 |
| Δ-LFM | w/ Rank Loss | [0, 1] | ✓ | 28.36 ± 1.21 | 91.15 ± 1.11 | 0.272 ± 0.285 | 0.474 ± 0.088 |
| | w/ ArcRank Loss | [0, 1] | ✓ | 29.83 ± 0.95 | 92.74 ± 0.93 | 0.243 ± 0.277 | 0.454 ± 0.088 |
| | w/ ArcRank Loss | [0, T] | ✓ | **30.04 ± 1.04** | **92.63 ± 1.03** | **0.233 ± 0.282** | **0.442 ± 0.087** |

shows error in the ventricles; however, these are less intense, with thinner boundaries and smaller cortical-surface errors, demonstrating improved overall accuracy.

We report PSNR and SSIM in Table 1, and Region MAE together with $\Delta$-RMAE in Table 2. Across all three datasets, our proposed $\Delta$-LFM consistently achieves high performance.

In Table 1, $\Delta$-LFM improves PSNR by a clear margin over the strongest baseline. On ADNI, it reaches 30.59 dB, exceeding the second-best method (MambaControl, 29.72 dB) by +0.87 dB. Similar gains are observed on AIBL (+0.66 dB) and OASIS (+0.77 dB). The SSIM results mirror this trend: $\Delta$-LFM achieves 94.62, 93.92, and 89.36, improving upon the best baselines by roughly +1.0, +0.7, and +1.1 points, respectively. These consistent improvements indicate that $\Delta$-LFM produces reconstructions with both higher fidelity and sharper structural alignment.

Table 2 further validates these observations with volumetric error metrics. $\Delta$-LFM yields the lowest Region MAE on ADNI (0.210) and OASIS (0.262), and matches the best-performing method (DiffuseMorph) on AIBL (0.226). More importantly, it achieves the lowest $\Delta$-RMAE across all datasets: 0.436 on ADNI, 0.417 on AIBL, and 0.473 on OASIS. Compared to the strongest baseline (MambaControl), this represents relative error reductions of about 21%, 21%, and 16%, respectively.

## 4.3 ABLATION STUDY

We provide visualization results of the learned latent space in Figure 3 on test cases in ADNI dataset, where t-SNE is used to project the representations into two dimensions. As shown in the left panel, the latent trajectories of individual patients are well aligned. Interestingly, the right panel reveals that the latent space also organizes according to diagnostic status, despite the fact that this information was never used during training.

To assess the contribution of each component in our framework, we conducted an ablation study across three datasets, as summarized in Table 3. The experiments examine the effect of (i) introducing conditional information, (ii) different auto-encoder training strategies (Arc Loss, Rank Loss,

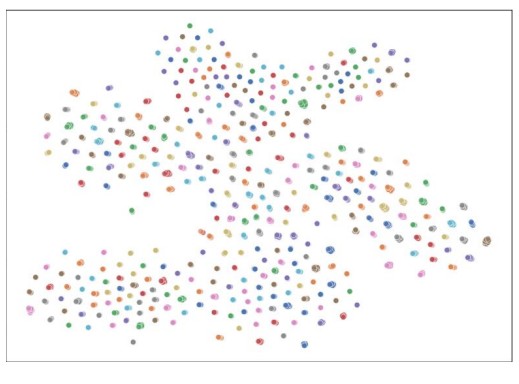
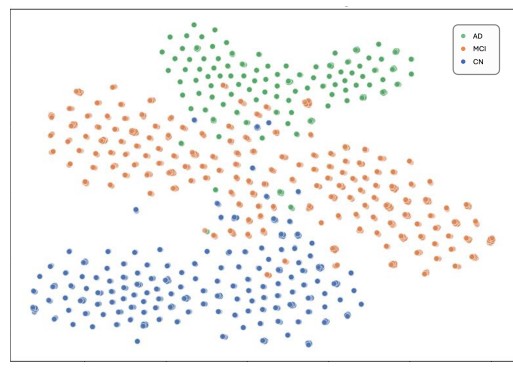

(a) colored by patient identity                    (b) colored by diagnosis status

Figure 3: t-SNE projection of the learned longitudinal latent space into two dimensions. (a) Coloring by subject ID shows that scans from the same patient cluster together. (b) Coloring by disease status shows that scans with the same diagnosis are grouped closely. CN: Cognitively Normal. MCI: Mild Cognitive Impairment. AD: Alzheimer's Disease (Dementia Stage).

Table 4: Ablation on the proposed components.

| Angular loss | Ranking loss | Sampling | PSNR | SSIM |
|---|---|---|---|---|
| Arc Loss | – | $[0, T]$ | 29.88 | 92.31 |
| – | Rank Loss | $[0, T]$ | 28.85 | 91.10 |
| Cosine Similarity | – | $[0, 1]$ | 29.13 | 91.10 |
| Cosine Similarity | Simple Rank Loss | $[0, 1]$ | 28.32 | 89.92 |
| Arc Loss | Simple Rank Loss | $[0, 1]$ | 28.44 | 89.87 |

ArcRank Loss), and (iii) different flow-matching (FM) manifold sampling strategies, namely physical sampling $[0, 1]$ versus temporal progression modeling $[0, T]$.

The baseline Latent Flow Matching (LFM) (Dao et al., 2023) model without conditioning performs the worst across all metrics, showing notably low SSIM/PSNR and high error values. Incorporating conditional information (Conditional LFM) significantly improves image quality (SSIM from 88.62 to 91.20; PSNR from 27.59 to 28.46) and reduces both Region MAE and $\Delta$-RMAE.

Within the $\Delta$-LFM ablation, we observe two consistent trends. First, the choice of loss function has a direct impact: using Arc Loss alone yields clear improvements, whereas Rank Loss alone slightly undermines performance. This is because Arc Loss encourages patient trajectories to follow a meaningful direction, while Rank Loss without Arc imposes only a weak magnitude constraint. In contrast, their combination (ArcRank Loss) achieves the best balance, suggesting that structural alignment (Arc) and temporal ordering (Rank) are complementary. Second, extending the FM sampling strategy from $[0, 1]$ to $[0, T]$ further improves results by promoting gradual residual modeling over meaningful temporal intervals.

Finally, we visualize qualitative progression predictions in Figure 4 using cases from three datasets, demonstrating the ability of $\Delta$-LFM to predict future timepoints at arbitrary intervals. The results show disease-related deformations primarily affecting the ventricles and surrounding cortical areas, which is consistent with clinical observations.

We present qualitative visualizations of predicted disease progression in Figure 4. Each row illustrates disease trajectories over nine years generated by $\Delta$-LFM. The intermediate steps provide clinically interpretable insights into the progression process, capturing characteristic neurodegenerative patterns such as ventricular enlargement and cortical thinning in the temporal and parietal regions. These deformations align with established clinical observations, underscoring the biological plausibility of our model.

To further verify each proposed component, we provide additional ablations in Table 4. We evaluate cosine similarity $\mathcal{L}_{\text{cosine}}$ as a surrogate for Arc loss and a simple ranking loss $\mathcal{L}_{\text{simple}}$ as a surrogate

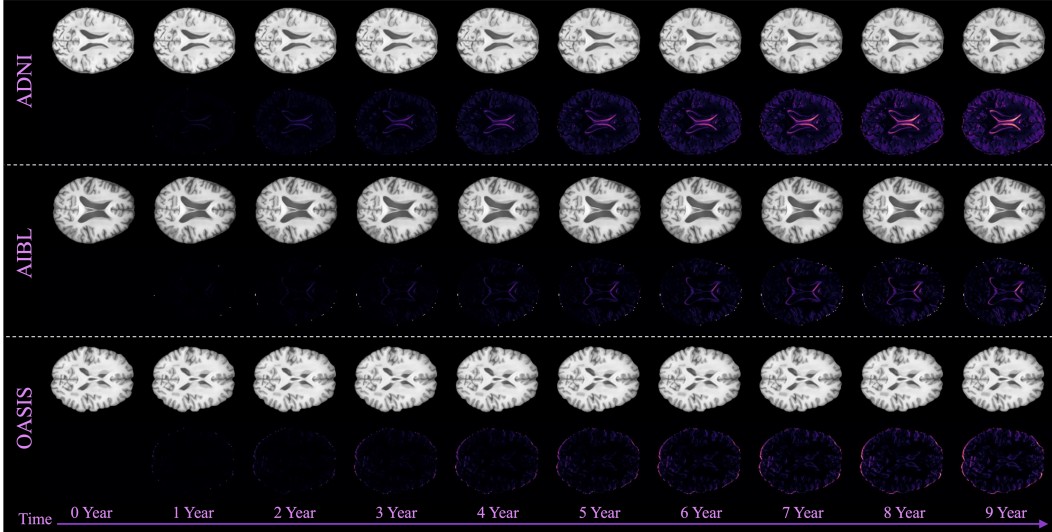

Figure 4: Continuous disease trajectory predicted by $\Delta$-LFM at one-year intervals. Odd rows show the baseline MRI (year 0) and predicted MRIs from years 1–9; even rows show residual maps relative to baseline. Progression highlights structural abnormalities, primarily in atrophy-related regions (*e.g.*, hippocampus, ventricles, surrounding cortex).

for Rank loss, defined as follows:

$$\mathcal{L}_{\text{cosine}}(i, j) = 1 - \frac{\langle \mathbf{z}_i, \mathbf{z}_j \rangle}{\|\mathbf{z}_i\| \, \|\mathbf{z}_j\|}, \qquad \mathcal{L}_{\text{simple}}(i, j) = \max\big(0, \, d_j - d_i\big). \tag{18}$$

Cosine similarity performs slightly worse than Arc loss, which is expected since it only enforces a global angular alignment and does not capture the fine-grained latent directional structure modeled by Arc loss. In contrast, the simple rank loss without a margin often leads to latent mode collapse, explaining its poor performance when combined with either Arc loss or cosine similarity.

**Loss Weight.** $\lambda_{\text{arc}}$ is relatively sensitive. Values $\lambda_{\text{arc}} > 0.01$ consistently cause collapse, while $0.0025$–$0.0075$ are stable across runs, with $\lambda_{\text{arc}} = 0.0075$ noticeably degrading AE reconstruction. We therefore set $\lambda_{\text{arc}} = 0.005$ as a compromise, trading a small decrease in AE reconstruction for improved flow-matching performance ($+0.4$ PSNR over $0.0025$).

For $\lambda_{\text{rank}}$, values above $0.05$ hinder convergence, whereas the range $0.005$–$0.02$ yields similar behavior. We choose $\lambda_{\text{rank}} = 0.01$, which gives a slight PSNR gain ($+0.05$).

**Speed Analysis** ArcRank involves an SVD operation, which is computational expensive. In practice, we use `torch.linalg.svd(z, full_matrices=False)`, which gives a $6\times$ speedup compared to the full matrices variant ($0.0552$s $\rightarrow 0.0092$s on average over 50 runs). Overall, ArcRank increases AE training time by $\sim 40\%$ ($1.4\times$), which we regard as a reasonable overhead given the resulting performance gains.

## 5 CONCLUSION

We introduce **Progression Latent Flow Matching ($\Delta$-LFM)**, a framework that unifies the proposed ArcRank loss with temporal flow matching to capture patient-specific disease trajectories. To evaluate progression in isolation, we propose a new metric, $\Delta$-RMAE. On longitudinal MRI benchmarks, $\Delta$-LFM consistently achieves higher fidelity, lower error, and tighter alignment with ground-truth progression than prior methods. Beyond accuracy, it yields temporally coherent and clinically interpretable trajectories. These results show that $\Delta$-LFM provides a new direction for progression modeling, with the potential to generalize across other temporal generation tasks.

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

# Appendix

## A  TRAINING SETTING

We adopt the AdamW optimizer (Loshchilov & Hutter, 2017) for all experiments. For the autoencoder (AE) and 3D U-Net (Ronneberger et al., 2015), we trained with a learning rate of $1 \times 10^{-3}$ and a batch size of 2 for 300 epochs. For the flow matching model based on the 3D U-Net, we used a reduced learning rate of $3 \times 10^{-5}$ with a larger batch size of 4 for 200 epochs.

## B  CONDITIONAL GENERATION

Following Litrico *et al.* (Litrico et al., 2024), we encode the current sample time in flow matching into continuous embeddings using sinusoidal positional encodings. These time embeddings act as an additive bias to the input, ensuring that the generative process is aware of its temporal position.

Building on this, and inspired by BrLP (Puglisi et al., 2024), we introduce *explicit control signals* that combine patient-specific attributes with temporal queries. In particular, we condition the model not only on the start time, the current sample time, and the end query time, but also on the patient attributes. This design enables the model to incorporate both subject-specific and task-specific information when shaping the generative trajectory.

To inject these conditional signals into the U-Net backbone, we explored three strategies: (1) Additive biasing, (2) Cross-attention (Vaswani et al., 2017) and (3) Adaptive Layer Normalization (AdaLN) (Raffel et al., 2020).

Among these approaches, our experiments show that AdaLN is most effective with our temporal sampling over the interval $[0, T]$. It adapts the network representation in a temporally aware and context-sensitive manner, allowing the generative process to be steered smoothly by patient attributes and target query times.

The implementation of AdaLN is realized through an initial one-layer MLP that processes the condition signals, followed by additional MLPs at each U-Net decoder layer to generate the corresponding AdaLN modulation parameters.

## C  AUTO-ENCODER ABLATION

Table 5 summarizes the effect of AE capacity and training crop size. For the capacity ablation, we vary the encoder channels while fixing the crop size to $64^3$. For the crop-size ablation, we fix the AE to $[64, 128, 256]$ and compare $48^3$ vs. $64^3$ crops.

Table 5: Ablation on AE capacity and training crop size.

| AE capacity (crop $64^3$) | | | Crop size (AE $[64, 128, 256]$) | | |
|---|---|---|---|---|---|
| **Channels** | **PSNR** | **SSIM** | **Crop** | **PSNR** | **SSIM** |
| $[64, 128, 256]$ | 30.04 | 92.63 | $64^3$ | 30.04 | 92.63 |
| $[64, 128, 128]$ | 29.89 | 92.55 | $48^3$ | 29.51 | 91.71 |
| $[64, 128]$ | 29.62 | 92.10 | | | |

We do not use deeper or wider AEs, as these configurations lead to out-of-memory (OOM) errors on our RTX A6000 GPU (48 GB). Based on the above results, we use channels $[64, 128, 256]$ and crop size $64^3$ as our default configuration. Larger crops (e.g., $72^3$) also result in OOM on our hardware. To keep the comparison with baselines fair, we do not use a pretrained AE, although initializing with MAISI (Guo et al., 2025) can further improve PSNR (from 30.04 to 31.97 for $[64, 128, 256]$).

Table 6: Quantitative results across different prediction horizons (years into the future). Measurements are reported using PSNR (dB), SSIM ($\times$1.0E2), Region MAE, and $\Delta$-RMAE.

| Year | PSNR $\uparrow$ | SSIM $\uparrow$ | Region MAE $\downarrow$ | $\Delta$-RMAE $\downarrow$ |
|------|-----------------|-----------------|-------------------------|----------------------------|
| 1 | $31.61 \pm 0.87$ | $93.08 \pm 0.75$ | $0.209 \pm 0.107$ | $0.386 \pm 0.107$ |
| 2 | $31.79 \pm 0.92$ | $93.79 \pm 0.77$ | $0.217 \pm 0.106$ | $0.369 \pm 0.097$ |
| 3 | $31.09 \pm 0.95$ | $93.51 \pm 0.82$ | $0.206 \pm 0.118$ | $0.388 \pm 0.089$ |
| 4 | $32.48 \pm 0.93$ | $93.95 \pm 0.89$ | $0.201 \pm 0.132$ | $0.395 \pm 0.084$ |
| 5 | $32.14 \pm 0.98$ | $93.59 \pm 0.88$ | $0.215 \pm 0.148$ | $0.405 \pm 0.080$ |
| 6 | $31.37 \pm 1.02$ | $93.07 \pm 0.95$ | $0.224 \pm 0.163$ | $0.414 \pm 0.081$ |
| 7 | $30.12 \pm 0.98$ | $92.58 \pm 1.07$ | $0.241 \pm 0.176$ | $0.436 \pm 0.082$ |
| 8 | $29.83 \pm 1.05$ | $92.18 \pm 1.15$ | $0.261 \pm 0.191$ | $0.459 \pm 0.089$ |
| 9 | $28.09 \pm 1.07$ | $90.65 \pm 1.20$ | $0.279 \pm 0.207$ | $0.487 \pm 0.085$ |
| 10 | $28.61 \pm 1.23$ | $91.41 \pm 1.37$ | $0.302 \pm 0.246$ | $0.552 \pm 0.104$ |
| 11 | $28.71 \pm 1.13$ | $91.32 \pm 1.22$ | $0.294 \pm 0.218$ | $0.506 \pm 0.094$ |
| 13 | $26.98 \pm 1.29$ | $89.96 \pm 1.44$ | $0.339 \pm 0.259$ | $0.579 \pm 0.109$ |

## D   LONGITUDINAL PROGRESSION QUANTITATIVE RESULTS

Table 6 summarizes predictive performance across different future horizons. For short- to mid-term forecasts (1–5 years), reconstruction metrics remain consistently high: PSNR hovers around 31–32 dB and SSIM above 93%, indicating strong fidelity in reproducing structural details. Region MAE is also relatively stable in this range ($\sim 0.20$), suggesting reliable local accuracy, while $\Delta$-RMAE values remain below 0.41, reflecting well-preserved progression dynamics.

Beyond 6 years, however, all metrics exhibit gradual degradation. PSNR and SSIM decline steadily, dropping to 28 dB and below 91% by year 10, and further to 26.98 dB / 89.96% at year 13. Concurrently, Region MAE and $\Delta$-RMAE increase monotonically, indicating that both absolute errors in regional values and their temporal progression deviate more strongly with longer horizons. Notably, the sharp rise in $\Delta$-RMAE after year 10 ($> 0.50$) highlights a growing difficulty in capturing long-term progression trends.

We also observe that the prediction trend is not strictly linear as the prediction horizon increases. This is mainly because different time horizons are estimated from different subsets of subjects. In particular, fewer subjects have follow-up data at 10–13 years compared to 1–5 years, which introduces higher variability and heterogeneity, leading to slightly inconsistent trends.

## E   ON THE STABILITY OF $\Delta$-RMAE

Residual-based metrics such as $\Delta$-RMAE can in principle be affected by imperfect registration and intensity inhomogeneity, and our preprocessing cannot guarantee perfectly aligned, bias-free images. Here we analyze how such imperfections influence the metric.

In our setting, $\Delta$-RMAE is defined voxel-wise as

$$\Delta\text{-RMAE} = \frac{\left|\Delta_{\text{gt}} - \Delta_{\text{gen}}\right|}{\frac{1}{2}\left(|\Delta_{\text{gt}}| + |\Delta_{\text{gen}}|\right)}, \tag{19}$$

so that $\Delta$-RMAE $\in [0, 2]$, with values close to 2 indicating large disagreement between the two residuals. We model misregistration and residual bias-field effects as an additive perturbation $\Delta_{\text{mis}}$ on the ground-truth residual, yielding

$$\Delta\text{-RMAE}_{\text{noise}} = \frac{\left|\Delta_{\text{mis}} + \Delta_{\text{gt}} - \Delta_{\text{gen}}\right|}{\frac{1}{2}\left(|\Delta_{\text{mis}} + \Delta_{\text{gt}}| + |\Delta_{\text{gen}}|\right)}. \tag{20}$$

When $\Delta_{\text{mis}}$ is small relative to $\Delta_{\text{gt}}$ and $\Delta_{\text{gen}}$, numerator and denominator are perturbed in a comparable way, so $\Delta$-RMAE$_{\text{noise}}$ remains close to the ideal $\Delta$-RMAE. As $\Delta_{\text{mis}}$ increases, the main effect is a mild compression of the dynamic range (the effective upper bound becomes slightly below 2), rather than a systematic bias toward any particular model.

To quantify this effect, we perform a simple sensitivity analysis by adding Gaussian noise with standard deviation $\sigma$ to the residuals and recomputing $\Delta$-RMAE. Table 7 reports the mean, standard

Table 7: Sensitivity of $\Delta$-RMAE to additive Gaussian noise on the residuals.

| $\sigma$ (noise) | Mean $\Delta$-RMAE | Standard Deviation | Bias vs. True |
|---|---|---|---|
| 0.00 | 2.000000 | 0.000000 | 0.000000 |
| 0.05 | 1.999842 | 0.002315 | -0.000158 |
| 0.10 | 1.999375 | 0.005982 | -0.000625 |
| 0.15 | 1.998214 | 0.011430 | -0.001786 |
| 0.20 | 1.996102 | 0.019752 | -0.003898 |
| 0.25 | 1.992994 | 0.031245 | -0.007006 |
| 0.30 | 1.989124 | 0.046521 | -0.010876 |
| 0.35 | 1.983612 | 0.064210 | -0.016388 |
| 0.40 | 1.976348 | 0.085402 | -0.023652 |
| 0.45 | 1.968740 | 0.107895 | -0.031260 |
| 0.50 | 1.960255 | 0.133005 | -0.039745 |
| 0.55 | 1.950492 | 0.160884 | -0.049508 |
| 0.60 | 1.939101 | 0.191156 | -0.060899 |
| 0.65 | 1.926345 | 0.222974 | -0.073655 |
| 0.70 | 1.911890 | 0.255780 | -0.088110 |
| 0.75 | 1.896232 | 0.288901 | -0.103768 |
| 0.80 | 1.879110 | 0.322410 | -0.120890 |
| 0.85 | 1.861775 | 0.355122 | -0.138225 |
| 0.90 | 1.843569 | 0.386230 | -0.156431 |
| 0.95 | 1.834021 | 0.415104 | -0.165979 |
| 1.00 | 1.826570 | 0.462351 | -0.173430 |

deviation, and bias relative to the noiseless value (2.0). Even for relatively large noise levels ($\sigma$ up to 1.0), the induced bias remains small compared to the full range $[0, 2]$, supporting the robustness of $\Delta$-RMAE to moderate misregistration.

## F CLINICAL CONDITIONING: SETUP AND VERIFICATION

We conduct a series of analyses to assess the robustness of the model to perturbations in the conditioning metadata.

**Age.** Age serves as the temporal anchor for longitudinal prediction, as future timepoints are defined by (*baseline age + time span*). Injecting noise into age would therefore shift the implied prediction timepoint, so that the generated image no longer corresponds to a meaningful clinical follow-up. Such an ablation would be difficult to interpret, and for this reason we do not treat age as a noise-robust attribute.

**Sex.** To evaluate robustness with respect to sex, we intentionally flipped the sex attribute and re-evaluated the model. On ADNI, performance changed only marginally, from PSNR/SSIM of $30.59/94.62$ to $30.12/94.47$, indicating limited sensitivity to this source of label noise.

**Clinical status.** In clinical practice, diagnostic labels are inherently noisy. To reflect this, we inject Gaussian noise into the clinical status during training to model diagnostic uncertainty. Specifically, we use zero-mean Gaussian noise with standard deviation equal to one-third of the status class interval (interval $= 3$).

Under perturbed status labels, performance remains reasonable, with PSNR/SSIM degrading only slightly from $30.59/94.62$ to $29.81/93.58$, demonstrating robustness to clinically realistic label noise.

## G LIMITATIONS

Our work, while demonstrating the potential of latent flow matching for longitudinal imaging generation, has several limitations that we wish to acknowledge.

**Task-specific limitation.** In the current study, we restrict our focus to AD, which provides one of the most widely available longitudinal imaging datasets in the medical domain. This restriction is

pragmatic, as AD offers abundant imaging data suitable for model training and evaluation. However, it inevitably narrows the generalizability of our findings. In future work, we aim to extend our framework to other progressive diseases such as brain tumors, where longitudinal imaging carries unique challenges, including rapid progression patterns, heterogeneous lesion appearance, and treatment-induced alterations. Moreover, we encourage exploration of our method beyond medical imaging, where modeling temporal dynamics may shed light on broader scientific and engineering problems.

**Dataset processing limitation.** Although we followed standard MRI preprocessing protocols, variations across scanners and acquisition protocols introduce substantial heterogeneity in the data. Such differences may affect the stability of model training and the reliability of generated outputs. Furthermore, we identified occasional metadata inconsistencies (*e.g.*, mislabeling of imaging orientation), which led to registration failures in certain cases. While these issues were infrequent, they highlight the importance of rigorous data curation and quality control in longitudinal imaging research. Future work may benefit from harmonization techniques or scanner-invariant representations to further mitigate such variability.

## H  FUTURE WORK

In future research, we plan to enrich our framework in several important directions.

First, the incorporation of explicit anatomical information could enable more fine-grained modeling of disease progression. By constraining generation with structure-aware priors (e.g., hippocampus, ventricles, cortical regions), the model may better capture local pathological changes while preserving biological plausibility.

Second, beyond modeling each patient in isolation, future work may explore leveraging inter-patient disease trend similarity and dissimilarity. Identifying cohorts of patients with comparable progression patterns may enable transfer of knowledge across individuals, while simultaneously capturing unique patient-specific deviations. In addition, modeling nonlinear progression dynamics remains an important direction for future work.

Addressing dataset heterogeneity remains another critical challenge. Future works may investigate harmonization techniques and scanner-invariant representations to mitigate variability introduced by different acquisition protocols and hardware. Such advances will be essential for scaling the framework to large, multi-center cohorts and for ensuring robust generalization in real-world clinical applications.

## I  DISCUSSION OF LLM USAGE

We used large language models (LLMs), including ChatGPT (OpenAI, 2023) and Gemini (Team et al., 2023), as writing assistants for grammar correction and style refinement. All technical content, methodology, and experiments were developed by the authors.

