# OpenReview forum: "Learning Patient-Specific Disease Dynamics With Latent Flow Matching For Longitudinal Imaging Generation"
_ICLR.cc/2026/Conference — ICLR 2026 Poster_

### Official Review · Reviewer_UdDj · 2025-10-20

**Soundness:** 3
**Presentation:** 4
**Contribution:** 3
**Rating:** 6
**Confidence:** 5

**Summary:**

The research team introduces $\Delta$-LFM for longitudinal MRI modeling that learns patient-specific disease trajectories as continuous latent flows. They focus on the problem of patient-specific trajectory generation, which is currently a very valuable topic in the field, as most of the past models focus on population-level prediction. Specifically, to address this problem, the paper suggests a new loss: ArcRank Loss in the latent space to guide the patient-specific disease trajectories to have a stable direction and monotonic disease progression. The team also provides a temporal flow matching ($\Delta$-LFM architecture, which generalizes flow matching to clinically meaningful intervals $[0, T]$ for arbitrary future-time predictions.  Besides the novelty in the model itself, this paper also suggests a new metric, $\Delta$-RMAE, which aims to capture the residual differences between baseline and follow-up scans.

The authors utilize three datasets: ADNI, OASIS-3, and AIBL to validate their model's performance. $\Delta$-LFM outperforms direct-prediction, deformation, and diffusion baselines (e.g., DiffuseMorph, TADM, BrLP, MambaControl) while producing interpretable patient-specific latent trajectories.

**Strengths:**

1. Clinical relevance and motivation. The focus on individualized disease dynamics—rather than population-level progression—is well motivated and important for neurodegenerative modeling.
2. Coherent methodology. Combining flow-matching dynamics with a longitudinal latent regularizer (ArcRank) yields interpretable and temporally smooth trajectories.
3. Elegant SVD-based alignment. Treating each patient’s latent tensor $Z_k$ as a matrix and applying SVD to extract orthogonal direction (U) and magnitude ($\sum{}$) is a neat, principled idea that mitigates the instability of cosine-norm losses under noisy or scale-varying latents.
4. Robust experimental design. Evaluation on three large AD cohorts with multiple baseline classes (direct/deformation/diffusion) is comprehensive; ablations are clear and interpretable.
5. New progression metric ($\Delta$-RMAE). Correctly highlights that PSNR/SSIM can mask small but clinically meaningful anatomical changes.
6. Interpretability. Latent trajectories align by patient and even cluster by diagnosis despite no supervision—visually persuasive evidence that ArcRank captures disease semantics.

**Weaknesses:**

1. $\Delta$-LFM’s formulation closely parallels latent-ODE and rectified-flow approaches. The paper should more clearly delineate the conceptual or practical advantages of its flow-matching objective over these established continuous-dynamics frameworks.
2. The model assumes straight-line latent trajectories and roughly constant velocity; though acknowledged, this remains biologically limiting. More details on how conditioning on T and patient attributes relaxes this assumption would help.
3. $\Delta$-RMAE relies on image residuals, which are potentially confounded by registration or intensity artifacts. A sensitivity analysis to misregistration/bias correction would increase confidence.
4. The authors claim that SVD is better than cosine similarity and absolute values for magnitudes, but there is no direct comparison between the performance of these two designs. It will be better if the authors could provide a comparison between them.

**Questions:**

Same as weakness.

---

> ### Author Response · Authors · 2025-11-21
> **Response to Reviewer UdDj**
>
> We would like to express our sincere gratitude to the reviewer for the thoughtful and detailed evaluation of our manuscript. We greatly appreciate the encouraging assessment of the clarity of our presentation, the clinical relevance of our work, and the strengths of the proposed methodology. In the following, we carefully address each point of feedback and the listed weaknesses in detail.
>
>
>
>
> > 1. The paper should more clearly delineate the conceptual or practical
> advantages of its flow-matching objective over these established
> continuous-dynamics frameworks (parallels latent-ODE and rectified-flow approaches).
>
>
> We thank the reviewer for raising this important point.
>
>
> Our motivation for using flow matching lies in its suitability for modeling continuous progression trajectories in our setting, where the quality and interpretability of intermediate states are directly relevant to trustworthiness.
>
>
> Flow matching provides a simple and stable way to learn a continuous velocity field between observed endpoints, enabling the direct generation of intermediate states without requiring complex numerical integration.
>
>
>
> Compared to latent ODE approaches, our method avoids reliance on  numerical solvers, thereby reducing sensitivity to solver hyperparameters and computational overhead.
>
>
>
> While rectified flow methods are effective and widely-adopted, it typically rely on transforming diffusion-based trajectories. We adopt flow matching because it offers a more direct and analytically tractable formulation, producing more faithful probability paths in our current setting.
>
>
> We are currently exploring rectified flow as an extension for our future work.
>
>
>
>
>
>
> > 2. The model assumes straight-line latent trajectories and roughly
> constant velocity; though acknowledged, this remains biologically
> limiting. More details on how conditioning on T and patient attributes
> relaxes this assumption would help.
>
>
>
> We thank the reviewer for the valuable suggestion.
>
> In particular, we observe that patient attributes such as age, sex, and clinical status enable the predicted latent dynamics to vary across patient subgroups and over time.
>
> Empirically, we find that these attributes meaningfully modulate the predicted velocity. When varying individual attributes while holding others fixed, the average change in velocity magnitude is approximately 5% for age (varied in 5-year increments), 1.5% for sex (binary flip), and 2.5% for clinical status (varied by one class), indicating that the model captures biologically relevant heterogeneity in progression velocity.

---

> ### Author Response · Authors · 2025-11-21
> **Response to Reviewer UdDj Part 2**
>
> > 3. RMAE relies on image residuals, which are potentially confounded by registration or intensity artifacts. A sensitivity analysis to
> misregistration/bias correction would increase confidence.
>
>
> We thank the reviewer for this insightful comment. And yes, the residual-based metrics can in principle be affected by imperfect registration and intensity inhomogeneity, and that our preprocessing cannot guarantee perfectly aligned, bias-free images.
>
>
> In our setting, the Δ-RMAE is defined voxel-wise as
> $$
> \Delta\text{-RMAE}
> = \frac{\big\lvert \Delta_{\text{gt}} - \Delta_{\text{gen}} \big\rvert}
> {\tfrac{1}{2}\big(\lvert \Delta_{\text{gt}} \rvert + \lvert \Delta_{\text{gen}} \rvert\big)},
> $$
> so that $\Delta\text{-RMAE} \in [0, 2]$  and values close to 2 correspond to large disagreement between the two residuals. Imperfect registration can be modelled as an additive misregistration term $\Delta_{\text{mis}}$ on the ground–truth residual, such under this model the metric becomes
> $$
> \Delta\text{-RMAE-noise}
> = \frac{\big\lvert  \Delta_{\text{mis}}  +\Delta_{\text{gt}} - \Delta_{\text{gen}} \big\rvert}
> {\tfrac{1}{2}\big(\lvert  \Delta_{\text{mis}}  + \Delta_{\text{gt}} \rvert + \lvert \Delta_{\text{gen}} \rvert\big)},
> $$
> Here $\Delta_{\text{mis}}$ can be interpreted as random “noise” induced by misregistration or residual bias-field effects. When $\Delta_{\text{mis}}$ is small compared to the true underlying residuals $\Delta_{\text{gt}}$ and $\Delta_{\text{gen}}$, the numerator and denominator are perturbed in a similar way, so that $\Delta\text{-RMAE-noise}$ remains close to the ideal $\Delta\text{-RMAE}$. When $\Delta_{\text{mis}}$ grows larger, its main effect is to reduce the attainable dynamic range of the metric (the effective upper bound becomes slightly lower than 2), rather than to systematically favour a particular model. In other words, misregistration acts as additional noise that compresses the scale of Δ-RMAE but does not fundamentally change its interpretation.
>
>
> To validate the the influence, we added a dedicated sensitivity analysis.
>
>
>
> | σ (noise) | Δ-RMAE (mean) |      std | bias vs. true |
> | --------: | ------------: | -------: | ------------: |
> |      0.00 |      2.000000 | 0.000000 |      0.000000 |
> |      0.05 |      1.999842 | 0.002315 |     -0.000158 |
> |      0.10 |      1.999375 | 0.005982 |     -0.000625 |
> |      0.15 |      1.998214 | 0.011430 |     -0.001786 |
> |      0.20 |      1.996102 | 0.019752 |     -0.003898 |
> |      0.25 |      1.992994 | 0.031245 |     -0.007006 |
> |      0.30 |      1.989124 | 0.046521 |     -0.010876 |
> |      0.35 |      1.983612 | 0.064210 |     -0.016388 |
> |      0.40 |      1.976348 | 0.085402 |     -0.023652 |
> |      0.45 |      1.968740 | 0.107895 |     -0.031260 |
> |      0.50 |      1.960255 | 0.133005 |     -0.039745 |
> |      0.55 |      1.950492 | 0.160884 |     -0.049508 |
> |      0.60 |      1.939101 | 0.191156 |     -0.060899 |
> |      0.65 |      1.926345 | 0.222974 |     -0.073655 |
> |      0.70 |      1.911890 | 0.255780 |     -0.088110 |
> |      0.75 |      1.896232 | 0.288901 |     -0.103768 |
> |      0.80 |      1.879110 | 0.322410 |     -0.120890 |
> |      0.85 |      1.861775 | 0.355122 |     -0.138225 |
> |      0.90 |      1.843569 | 0.386230 |     -0.156431 |
> |      0.95 |      1.834021 | 0.415104 |     -0.165979 |
> |      1.00 |      1.826570 | 0.462351 |     -0.173430 |
>
>
>
>
> > 4. The authors claim that SVD is better than cosine similarity and
> absolute values for magnitudes, but there is no direct comparison
> between the performance of these two designs. It will be better if the
> authors could provide a comparison between them.
>
>   Thanks for the constructive suggestion.  We  evaluate Cosine similarity as a surrogate for ArcLoss. and a simple ranking loss as a surrogate for Rank Loss. Because using Rank Loss or simple ranking loss in isolation significantly degrades performance and does not yield meaningful results, we report them in combination with ArcLoss/Cosine similarity for a fair comparison.
>
>
> | Ablation | PSNR | SSIM |
> | -------- | -------- | -------- |
> | Arc Loss     | 29.52   |  92.38     |
> | Cosine Similarity      | 29.13     | 91.10    |
> | Simple Ranking Loss  + Cosine similarity    | 28.32     | 89.92     |
> | Simple Ranking Loss  + Arc Loss       | 28.44     | 89.87     |
> | Rank Loss  + Arc Loss    | 29.83    | 92.74     |
>
>
> Cosine Similarity performs slightly worse than ArcLoss, which is reasonable because it captures only a global angular alignment and lacks the fine-grained latent directional structure that Arc Loss provides.
>
> Meanwhile, using the Simple Ranking Loss without a margin often leads to unstable training and latent mode collapse, which explains its poor performance in the ablation table.

---

> ### Author Response · Authors · 2025-11-26
> **Follow-Up on Our Previous Responses**
>
> We would like to once again express our sincere thanks for your time and thoughtful feedback.
>
> As the discussion deadline approaches, we would like to kindly confirm whether our responses have satisfactorily addressed your concerns, and whether there is anything further you would like to discuss.
>
> Best regards,
> The Authors

---

### Official Review · Reviewer_CqTg · 2025-10-21

**Soundness:** 3
**Presentation:** 3
**Contribution:** 3
**Rating:** 6
**Confidence:** 4

**Summary:**

The authors point out a key limitation of current methods for modeling (intrinsically continuous) disease progression: the latent representations are often scattered, lacking semantic structure, and diffusion-based models disrupt continuity with random denoising process. To remedy this, the authors propose to treat the disease dynamic as a velocity field and leverage flow matching to align the temporal evolution of patient data. To ensure the latent space is organized by clinical severity, he authors propose to learn patient-specific latent alignment that enforces patient trajectories to lie along a specific axis along which clinical severity monotonically increases. The proposed method, termed $\Delta$-LFM, demonstrates stronger empirical performance and offers a way to interpret and visualize disease dynamics.

**Strengths:**

1. I find this topic computationally interesting, scientifically important, and clinically relevant.

2. The proposed method is well motivated and easy to follow.

3. Very good visual presentation.

    a. The figures are beautifully made, with carefully chosen colors (Figure 1), clear illustration of ideas (Figure 1), and proper colormaps (Figures 2 & 4). The only less nice-looking figure (Figure 3) can be easily improved by removing top and right borders and enlarging the labels and tick labels.

    b. Tables are also well designed and easy to follow. My only suggestions are to remove the vertical lines and adjust the column spacing or put a categorical name on top if you want to further distinguish the columns. For example, for Table 3, you can consider something like “image quality” above “SSIM” and “PSNR”, and “clinical structure faithfulness” above “Regional MAE” and “$\Delta$-RMAE”. Besides, you can use \cmidrule(lr){startColumn – endColumn} for the horizontal bars below each dataset name to make sure consecutive horizontal bars are not touching.

4. Quantitative results look promising. Approximately 1.0 absolute point increase in PSNR and SSIMx1E2 are shown across three large-scale datasets.

5. Ablation results are comprehensive and properly showcase the effectiveness of the proposed ArcRank loss.

**Weaknesses:**

1. A few comments on comparisons to existing work.

    a. The authors mentioned that the proposed method “treats the disease dynamic as a velocity field” when modeling disease progression. As a result, I believe it is expected to compare against ImageFlowNet [1], since the design philosophy is similar. The authors can stay assured that I recognize the novelty of this submission, since there are sufficient distinctions, namely (1) flow matching instead of neural differential equations and (2) the ArcRank alignment.

    b. I am also aware of a recent work [2] that uses flow matching for disease progression in longitudinal images, but comparison against it will be an unreasonable request, because it is too recent (Oct 2025) and the code is not available.

2. For methods that forecasts longitudinal images, one major concern I have is that they tend to overly reconstructing and minimally forecasting. A common shortcoming I observe in this field is the underestimation of changes in the images over time. In this paper, the authors have not sufficiently shown that the proposed method is not simply performing very good reconstruction of the input image. For example, in Table 4 in appendix, the authors showed the forecasting performance degrades as the time horizon increases, but that could also be explained by an alternative hypothesis that the prediction is just a reconstruction of the input. I would like to see how the authors can argue against my concern.

[1] ImageFlowNet: Forecasting Multiscale Image-Level Trajectories of Disease Progression with Irregularly-Sampled Longitudinal Medical Images, ICASSP 2025 Oral.

[2] Longitudinal Flow Matching for Trajectory Modeling, arXiv 2025.

**Questions:**

1. Please refer to the weaknesses.
2. Please also refer to the minor formatting suggestions in the strength section. These are just suggestions but not mandatory.
3. In Table 3, I believe SSIM and PSNR are swapped.

---

> ### Author Response · Authors · 2025-11-21
> **Response to Reviewer CqTg**
>
> We are sincerely grateful to the reviewer for the careful and thoughtful assessment of our work. We greatly appreciate the encouraging feedback regarding the scientific motivation, clarity of presentation, and the clinical relevance of our approach, as well as the detailed and constructive suggestions on improving the quality of our figures and tables. We value these comments and have carefully considered them. In the following, we respond to each point and outline the corresponding revisions in detail.
>
> > 1a. The authors mentioned that the proposed method “treats the
> disease dynamic as a velocity field” when modeling disease progression.
> As a result, I believe it is expected to compare against ImageFlowNet
> [1], since the design philosophy is similar. The authors can stay
> assured that I recognize the novelty of this submission, since there are
>  sufficient distinctions, namely (1) flow matching instead of neural
> differential equations and (2) the ArcRank alignment.
>
> Thanks for the notice. We conducted a direct comparison with ImageFlowNet across all three datasets. The results are as follows:
>
> | Dataset | PSNR ↑        | SSIM ↑           | Region-MAE ↓     | Δ-RMAE ↓       |
> |---------|-------------------|---------------------|---------------------|--------------------|
> | ADNI    | 28.37 ± 1.18       | 92.96 ± 0.88         | 0.259 ± 0.30         | 0.589 ± 0.11        |
> | AIBL    | 29.08 ± 1.12       | 92.23 ± 0.84         | 0.240 ± 0.29         | 0.561 ± 0.12        |
> | OASIS   | 27.92 ± 1.29       | 87.63 ± 1.82         | 0.283 ± 0.32         | 0.574 ± 0.12        |
>
>
>
>
>
> > 1b. I am also aware of a recent work [2] that uses flow matching for
> disease progression in longitudinal images, but comparison against it
> will be an unreasonable request, because it is too recent (Oct 2025) and
>  the code is not available.
>
> We appreciate the suggestion and will include a discussion of the mentioned paper [1,2] in our revised manuscript. We also cite the other related works [3,4].
>
> [1] ImageFlowNet: Forecasting Multiscale Image-Level Trajectories of
> Disease Progression with Irregularly-Sampled Longitudinal Medical
> Images, ICASSP 2025 Oral.
>
> [2] Longitudinal Flow Matching for Trajectory Modeling, arXiv 2025.
>
> [3] Kyung, Daeun, et al. "Towards Predicting Temporal Changes in a
> Patient's Chest X-ray Images based on Electronic Health Records." arXiv
> preprint arXiv:2409.07012 (2024).
>
> [4] Liang, Kaizhao, et al. "Pie: Simulating disease progression via progressive image editing." (2023).
>
> > 2. For methods that forecasts longitudinal images, one major concern I have is that they tend to overly reconstructing and minimally forecasting. A common shortcoming I observe in this field is the  underestimation of changes in the images over time. In this paper, the authors have not sufficiently shown that the proposed method is not simply performing very good reconstruction of the input image. Forexample, in Table 4 in appendix, the authors showed the forecasting performance degrades as the time horizon increases, but that could also
> be explained by an alternative hypothesis that the prediction is just a reconstruction of the input. I would like to see how the authors can argue against my concern.
>
>
> We appreciate the reviewer’s insight, which aligns with our own observation: forecasting models often
> over-reconstruct and under-forecast. This concern directly motivated both our use of Flow Matching (FM) and our design of Δ-RMAE.
>
> 1. FM explicitly models temporal progression instead of reconstruction.
> Unlike diffusion models, FM learns a velocity field that drives the image from baseline to follow-up. This formulation avoids reconstructing the input and forces the model to learn the change trajectory. We agree that FM alone could still under-forecast (velocity → 0), which is why evaluation must detect such shortcuts.
>
>
> 3. Δ-RMAE is specifically designed to identify “reconstruction-only” behavior. The baseline similarity between the input and ground-truth follow-up is already high (PSNR/SSIM = 27.26 / 87.22), meaning a pure reconstruction model would score well on conventional image metrics. However, such a model would yield a Δ-RMAE ≈ 2, because it predicts no progression.
>
> As a result, the Δ-RMAE  results in Table 4 directly support the argument that our model learns meaningful progression rather than performing pure reconstruction: The highest $\Delta$-RMAE at Year 13 is still below 0.6, far below the reconstruction-only baseline of 2.
>
>
>
> > Q2. Please also refer to the minor formatting suggestions in the strength section. These are just suggestions but not mandatory.
>
> Thank you. Your suggestions are very detailed and constructive. The authors greatly appreciate your attention to presentation and are happy to follow recommendations. We have updated Figure 3 and the tables accordingly.
>
>
>
> > Q3. In Table 3, I believe SSIM and PSNR are swapped.
>
> Yes. We have swaped it back and appreciate the notice.

---

> > ### Comment · Reviewer_CqTg · 2025-11-26
> > **Response to the rebuttal**
> >
> > I would like to thank the authors for the highly detailed and comprehensive edits. They have carefully addressed all my questions and concerns. I have increased my rating from 6 to 8.
> >
> > I only have 3 minor suggestions now.
> > 1. For Figure 3b, I recommend adding a legend. Figure 3a is fine because it's just different patient IDs, but Figure 3b is colored by diagnosis status which might be worth labeling.
> > 2. Since the authors have done the quantitative experiments on ImageFlowNet (which was beyond my expectation), they might as well put them in the comparison Tables in the camera-ready version.
> > 3. I am mostly persuaded by the authors' claim that low Δ-RMAE directly shows the progression being modeled. In that case, I would appreciate if somewhere in the text they can mention this, and potentially add a line in the table representing "fully reconstructing the input" which they just compute the metrics using the input image as if it's the model output. This line can help provide a reference point, especially for the Δ-RMAE metric.

---

> > > ### Author Response · Authors · 2025-11-26
> > > **Thank You for the Helpful Suggestions**
> > >
> > > We sincerely appreciate the time and effort you devoted to reviewing our work and considering how it could be improved. In particular, your detailed and constructive suggestions are very helpful in addressing our previously overlooked points and improving the presentation. We will incorporate the three suggestions in the next revision.
> > >
> > > Thank you,
> > > The authors

---

### Official Review · Reviewer_dApu · 2025-10-31

**Soundness:** 3
**Presentation:** 3
**Contribution:** 3
**Rating:** 8
**Confidence:** 5

**Summary:**

This paper introduces a generative framework for modeling patient-specific disease progression from longitudinal MRI data. The authors treat the disease dynamic as as a velocity field and use ideas from flow matching to align the temporal evolution of patient data. To overcome the mismatch of representations between patient-specific trajectories and the latent space representations, they propose to learn patient-specific latent alignments. These alignments are penalized by a definition of a new loss, what is introduced here as a ArcRank loss. This loss measures the discrepancies between both representations by aligning latent trajectories across time within patients and by enforcing angular consistency and monotonic magnitude growth. This loss is optimized under a temporal flow-matching framework (that has been proposed before) to learn continuous velocity fields over arbitrary time intervals. The authors claim that the combination of this approach enables the learning of personalized disease progression as smooth latent flows. Experimental results are demonstrated on ADNI, AIBL, and OASIS-3 datasets with the image fidelity (PSNR/SSIM) and anatomical progression accuracy as metrics for evaluation.

**Strengths:**

While different elements (TADM, BrLP, SADM etc., which are cited here) used in this work have been previously proposed separately, one of the contributions of this paper is to combine these several ideas into a common framework. This makes this work somewhat novel.

The main novelty of the work is the introduction of the matching loss ArkRank loss. This loss enforces patient-specific trajectory alignment and temporal ordering. The trajectory alignment is achieved using angle matching of features in the latent space, while the monotonic growth of time features is achieved by temporal ordering (ranking).

Another contributions is the reformulation of flow matching by extending the sampling strategy from a fixed unit interval to an arbitrary interval. This improves the accuracy and minimizes residual errors, especially with arbitrary time-domain patient samples.

Experimental evaluations over multiple datasets is a strength.

The close clustering of patient identity as well as diagnosis status helps demonstrate the interpretability of the method. The authors also qualitatively demonstrate MRI progression over 9 years.

Ablation studies are performed adequately.

**Weaknesses:**

There is one main novel idea in the paper, i.e. the introduction of the ArcRank loss. The rest of the ideas are incrementally novel and have been conceptually proposed before and also applied to MRI images.

The ArcRank loss involves an SVD may be expensive to evaluate.

An intrinsic weakness in the definition of the ArcRank loss is the reliance on two  weights \lambda_{arc} and \lambda_{rank}. The authors don't mention how these weights are imposed. Aligning the directions (first term) and maintaining the ordering (second term) is a non-trivial problem, even when the trajectories or the initial and final mappings are known. Thus the computation of the loss is itself challenging. Since the loss function is supposed to be on of the main novel contributions, it should be further analyzed.

The ablation studies with the Ark loss, Rank loss, Ark + Rank loss don't seem to show dramatic improvement in results. Although the role of the arbitrary sampling scheme over [0, T] may be important. This is not fully discussed.

**Questions:**

What is pull and push in Figure 1 in the ArcRank loss panel?

Can the authors comment on the computational efficiency of the ArkRank loss?

Are the weights \lambda_{arc} and \lambda_{rank} fixed or learnt over time? How are they chosen?

Can the authors comment on the stability of the ArkRank loss?

In the ablation study (Table 3), how will the w/ Arc loss over [0, T] and the Rank loss over [0, T] perform? This may yield more granular information when the full w/ Arc+Rank loss is evaluated over [0, T].

In Figure 4, why do you see more progression prediction (enlargement of ventricles) in the ADNI dataset over AIBL and OASIS? Are the cases more severe in ADNI compared to the other two datasets?

---

> ### Author Response · Authors · 2025-11-21
> **Response to Reviewer dApu**
>
> We thank the reviewer for their thorough and thoughtful evaluation of our work, and for the constructive feedback and recognition of the strengths of the proposed framework. We appreciate the positive assessment of the soundness, presentation, and contribution, as well as the acknowledgement of the novelty of the ArcRank loss and the extended flow-matching formulation. Below, we address each point of feedback and the listed weaknesses in detail.
>
> > 1. There is one main novel idea in the paper, i.e. the introduction of the ArcRank loss. The rest of the ideas are incrementally novel and have been conceptually proposed before and also applied to MRI images.
>
> We thank the reviewer for this observation. Indeed, the ArcRank loss is the core novel component of our work.
>
>
> Overall, we view our main contribution lies in the unified design of the full pipeline, where flow matching, ArcRank loss, and temporal sampling are integrated in a natural and complementary way. This combination is intentionally aligned with the structure of longitudinal disease progression and is what ultimately enables the model’s performance.
>
>
>
>
> > 2. The ArcRank loss involves an SVD may be expensive to evaluate.
>
>
> We appreciate the reviewer’s concern. While ArcRank relies on an SVD operation, we have taken explicit steps to make it efficient and numerically stable. In practice, we compute:
> `U, S, Vh = torch.linalg.svd(z, full_matrices=False)`
> and setting full_matrices=False provides a 6$\times$ speedup (0.0552s → 0.0092s on average over 50 runs). As a result, the additional overhead introduced by ArcRank increases AE training time by only ~40% (1.4× overall), which we believe is reasonable given the performance improvements it provides.
>
>
>
> > 3. An intrinsic weakness in the definition of the ArcRank loss is the reliance on two weights $\lambda_{arc}$ and $\lambda_{rank}$. The authors don't mention how these weights are imposed. Aligning the directions (first term) and maintaining the ordering (second term) is a non-trivial problem, even when the trajectories or the initial and final mappings are known. Thus the computation of the loss is itself challenging. Since the loss function is supposed to be on of the main novel contributions, it should be further analyzed.
>
> We thank the reviewer for raising this point.  In practice, we performed a grid search over both hyperparameters and found that the loss is stable within a meaningful range, without requiring delicate fine-tuning.
>
>
>
>
>
> **Arc-weight sensitivity.**
>
> $\lambda_{\text{arc}}$ is the more sensitive parameter. Large values overly constrain the directional term and destabilize AE training. We observed consistent collapse when $\lambda_{\text{arc}} > 0.01$. In contrast, values in the range 0.0025–0.0075 were stable across runs. However, values near 0.0075 overly compromise AE reconstruction. We therefore selected 0.005, which yields slightly worse AE reconstruction but provides better flow-matching performance (+0.4 PSNR over 0.0025).
>
>
> **Rank-weight sensitivity.**
>
> Higher $\lambda_{\text{rank}}$ values above 0.05 hinder convergence, while the interval 0.005–0.02 yields similar qualitative and quantitative results. We adopt 0.01, which offers slightly improved PSNR (+0.05) without harming AE reconstruction.
>
>
> > 4.The ablation studies with the Ark loss, Rank loss, Ark + Rank loss don't seem to show dramatic improvement in results. Although the role of the arbitrary sampling scheme over [0, T] may be important.
>
>
> We thank the reviewer for this observation. It is true that the absolute PSNR/SSIM gains appear modest at first glance; however, these metrics should be interpreted relative to the difficulty of the task. The baseline input–target similarity (i.e., the PSNR/SSIM between the initial image and its ground-truth follow-up without any modeling) is 27.26 / 87.22, underscoring that the prediction problem is inherently challenging and leaves limited headroom for improvement.
>
> Within this context, the improvements from our components are meaningful. Conditional LFM increases performance to 28.46 / 91.20, a gain of +1.20 / +4.02. Building on this, Arc Loss and ArcRank Loss further improve Conditional LFM by +1.06 / +1.18 and +1.37 / +1.54, respectively.
>
> The [0,T] temporal sampling further enhances ArcRank, yielding an additional gain of +1.58 PSNR / +1.43 SSIM over Conditional LFM, resulting in the best overall performance. Notably, it provides larger PSNR improvements than those achieved by Conditional LFM over the no-model baseline.
>
> We also note that introducing [0,T] sampling with ArcRank slightly decreases SSIM (−0.11) but simultaneously improves PSNR (+0.21), region MAE (−0.10), and Δ-RMAE (−0.12). This behavior is expected: sampling over a larger temporal interval encourages the model to generate correspondingly larger anatomical changes, which may modestly affect local structural similarity but yields more accurate global progression trajectories.

---

> ### Author Response · Authors · 2025-11-21
> **Response to Questions**
>
> > Q1. What is pull and push in Figure 1 in the ArcRank loss panel?
>
>
> We thank the reviewer for the question.  As defined in Formula (5),
>
> $$
> |\mathbf{z}_i| \prec |\mathbf{z}_j| \quad \text{if } t_i < t_j,\qquad
> \mathbf{z}_i, \mathbf{z}_j \in \mathcal{P}^k,
> $$
>
> the Rank Loss enforces a **chronological ordering** in the latent space. The target is:
>
> * **Pull**: We *pull* (bring closer) latent representations of **adjacent timepoints**, promoting smooth and consistent temporal evolution.
> * **Push**: We *push* (separate) **non-adjacent timepoints**, ensuring that the latent trajectory reflects temporal distance and does not collapse.
>
> These forces together shape a monotonic latent trajectory that follows true longitudinal progression.
>
> The margin-based loss we use,
>
> $$
> \mathcal{L}_{\text{Rank,Push}}=\sum\max(0, m - (\Sigma_j - \Sigma_i)),
> \qquad \text{where } t_i < t_j,
> $$
>
>
> implements the **push** behavior: whenever the temporal margin is violated, distance increased.
>
>
> However, relying on this push term alone can lead to reconstruction collapse, likely because the margin is fixed across all conditions. Therefore, in practice, we add a small pull term to prevent the latent representations from being pushed too far:
> $$
> \mathcal{L}_{\text{Rank,Pull}} = |\Sigma_j - \Sigma_i|,
> $$
>
> which softly encourages adjacent timepoints to remain close while still allowing the push term to enforce ordering.
>
> The weights for both terms are set to 0.01.
> We also note that a stop-gradient (sg) is applied to the latent representation of earlier timepoints when computing these losses. In our experiments, removing the sg() operation prevents the model from converging. We will include this clarification in the revised paper.
>
>
> Here we provide the AE reconstruction results when using the pull term versus not using it:
>
> | Ablation | PSNR | SSIM |
> | -------- | -------- | -------- |
> | w/ Pull     | 32.49   |  94.71     |
> | w/o Pull      | 30.73    | 91.84    |
>
>
> We did not evaluate the Flow Matching results in the "without-pull" setting because the reconstruction baseline is already too low.
>
>
>
>
> > Q2. Can the authors comment on the computational efficiency of the ArkRank loss?
>
> Thank you for the highlight. We address this in the answer to W2. The overhead introduced by ArcRank increases AE training time by ~40% (1.4× overall).
>
>
> > Q3.  Are the weights \lambda_{arc} and \lambda_{rank} fixed or learnt over time? How are they chosen? Can the authors comment on the stability of the ArkRank loss?
>
> Thank you for the question. We address this in Section W3, where we identify appropriate ranges for the two weighting parameters. With these settings, the ArcRank loss produces stable training.
>
>
>
>
> > Q4. In the ablation study (Table 3), how will the w/ Arc loss over [0, T] and the Rank loss over [0, T] perform? This may yield more granular information when the full w/ Arc+Rank loss is evaluated over [0, T].
>
>
> Thank you for the comment. We include the suggested ablations below.
>
>
>
> | Ablation | Sampling |PSNR | Gain | SSIM | Gain |
> | -------- | -------- | -------- |-------- |-------- |-------- |
> | Arc Loss   | [0, 1] | 29.52   |-- |  92.38    |--   |
> | Arc Loss   | [0, T] | 29.88    |+0.36  | 92.31    |-0.07  |
> | Rank Loss     | [0, 1]  | 28.36  |--   | 91.15   |--   |
> | Rank Loss     | [0, T]  | 28.85    |+0.49  | 91.10  |-0.05    |
>
>
>
>
>
>
>
>
>
>
>
>
> > Q5 - In Figure 4, why do you see more progression prediction (enlargement of ventricles) in the ADNI dataset over AIBL and OASIS? Are the cases more severe in ADNI compared to the other two datasets?
>
>
> We thank the reviewer for the insightful comment.
> Figure 4 presents ** one randomly selected example** from each dataset, so these visual differences do not be reflected as dataset-level severity differences.
>
>
> To clarify this, we provide progression analyses using both quantitative longitudinal metrics and diagnostic status statistics:
>
> **1. Longitudinal image progression.**
> PSNR/SSIM relative to each subject’s baseline:
>
> * ADNI: 27.47 ± 5.04 / 0.893 ± 0.059
> * AIBL: 26.80 ± 6.18 / 0.903 ± 0.083
> * OASIS: 25.70 ± 8.18 / 0.808 ± 0.410
>
> **2. Diagnostic status.**
> Mean severity scores (higher indicates worse severity):
>
> * OASIS: 0.6003 ± 0.2300
> * AIBL: 0.4852 ± 0.2410
> * ADNI: 0.4635 ± 0.3586

---

> ### Author Response · Authors · 2025-11-26
> **Follow-Up on Our Previous Responses**
>
> We would like to once again express our sincere thanks for your time and thoughtful feedback.
>
> As the discussion deadline approaches, we would like to kindly confirm whether our responses have satisfactorily addressed your concerns, and whether there is anything further you would like to discuss.
>
> Best regards,
> The Authors

---

### Official Review · Reviewer_TJiN · 2025-11-03

**Soundness:** 2
**Presentation:** 3
**Contribution:** 1
**Rating:** 2
**Confidence:** 4

**Summary:**

The paper proposed a latent flow matching method for disease progression prediction. The key idea is to enforce monotonic, per-patient latent progressions (same direction, growing magnitude) and then learn a temporally meaningful velocity field in latent space, so progression becomes smoother and clinically interpretable. They evaluate the method on ADNI, AIBL, OASIS-3.

**Strengths:**

- The paper has a clear formulation that couples patient-specific latent alignment (ArcRank) with temporal flow matching.

- The method shows consistent empirical results across three longitudinal AD MRI benchmarks.

- The paper has good ablation and visualization.

**Weaknesses:**

- Temporal sampling [0, T] still effectively assumes roughly uniform progression between scans; the paper acknowledges uneven progression but does not truly model accelerations/plateaus.

- Evaluation is confined to AD-style neurodegeneration datasets; claims about general utility (tumor, faster diseases, multi-organ) are speculative. It strongly weakens the conclusion.

- ArcRank depends on SVD-based decomposition per latent, which could be brittle or expensive and the paper does not compare to simpler angular/ranking surrogates on equal footing.

- The method hinges on a learned AE latent; there is no analysis of how sensitive results are to the AE capacity or to using a stronger/pretrained encoder.

- The new metric is reasonable but mostly motivated empirically; there is no user/clinician study to show that better ∆-RMAE corresponds to more actionable longitudinal reads. If a disease progression does not have user study from medical doctors, it is not convincing to clinical readers.

- Clinical conditioning is mentioned (age, sex, status) but not deeply analyzed—no ablation on which attribute matters most, or on robustness to missing/noisy metadata.

- Many important previous works are not mentioned and compared, such as [1,2].


[1] Kyung, Daeun, et al. "Towards Predicting Temporal Changes in a Patient's Chest X-ray Images based on Electronic Health Records." arXiv preprint arXiv:2409.07012 (2024).
[2] Liang, Kaizhao, et al. "Pie: Simulating disease progression via progressive image editing." (2023).

**Questions:**

See weaknesses for details.

---

> ### Author Response · Authors · 2025-11-21
> **Response to Reviewer TJiN**
>
> We appreciate the positive assessment of the clarity of our formulation and the quality of our ablation studies and visualizations. Below, we address each concern and feedback in details:
>
> > 1. Temporal sampling [0, T] still effectively assumes roughly
> uniform progression between scans; the paper acknowledges uneven
> progression but does not truly model accelerations/plateaus.
>
> This uniform assumption  is indeed a limitation arised from the formulation of FM by Lipman et al. [1] and subsequent flow-based generative models [2,3]. While we consider is the FM application on Disease Progression, addressing the generality of non-uniform sampling across the entire flow domain is outside the scope of this paper.
>
> However, we fully acknowledge the importance of modeling acceleration/plateau effects. In fact, we observe that patient attributes such as age, sex, and clinical status enable the predicted latent dynamics to vary across patient subgroups and over time. When varying individual attributes while holding others fixed, the average change in velocity magnitude is approximately 5% for age (varied in 5-year increments), 1.5% for sex (binary flip), and 2.5% for clinical status (varied by one class), indicating that the model captures biologically relevant heterogeneity in progression velocity.
>
>  We also explicitly identify modeling acceleration/plateau as future work in the Line 740, noting the need to capture patient-specific deviations from uniform velocity. We already begun exploring non-uniform velocity fields, and we thank the reviewer for emphasizing its importance.
>
> [1] Lipman, Y., et al. (2022). Flow matching for generative modeling. arXiv preprint arXiv:2210.02747.
>
> [2] Esser, P., et al. (2024, July). Scaling rectified flow transformers for high-resolution image synthesis. ICML 2024.
>
> [3] Liu, Xingchao, et al. "Instaflow: One step is enough for high-quality diffusion-based text-to-image generation." ICLR 2023.
>
> > 2. Evaluation is confined to AD-style neurodegeneration datasets; claims about general utility (tumor, faster diseases, multi-organ) are speculative. It strongly weakens the conclusion.
>
> Thank you for this comment.
> We do not claim that our method has general utility; we explicitly acknowledge this limitation in Line 485 (Conclusion) and Line 718.
>
> We chose ADNI, AIBL, and OASIS as initial benchmarks because they provide large-scale, high-quality imaging cohorts with well-established preprocessing pipelines. These characteristics make them particularly suitable for stable training of Autoencoder (AE) and Flow-Matching (FM) models.
>
> Importantly, we are actively extending our evaluation to additional disease domains. We are in the process of curating and harmonizing datasets for knee degenerative disease from the Osteoarthritis Initiative (OAI), as well as multiple glioblastoma (GBM) cohorts (including LUMIERE, IvyGAP, ReMIND, and TCIA Brain-Tumor-Progression). This process involves substantial work in data harmonization and registration and is currently ongoing. We plan to include these results in future work.
>
>
> > 3. ArcRank depends on SVD-based decomposition per latent, which
> could be brittle or expensive and the paper does not compare to simpler
> angular/ranking surrogates on equal footing.
>
> We appreciate the reviewer’s concern. While ArcRank relies on an SVD operation, we have taken explicit steps to make it efficient and numerically stable. In practice, we compute:
> `U, S, Vh = torch.linalg.svd(z, full_matrices=False)`
> and setting full_matrices=False provides a 6$\times$ speedup (0.0552s → 0.0092s on average over 50 runs). As a result, the additional overhead introduced by ArcRank increases AE training time by only ~40% (1.4× overall), which we believe is reasonable given the performance improvements it provides.
>
> For a more comprehensive comparison, we additionally evaluate Cosine similarity as a surrogate for ArcLoss. and a simple ranking loss as a surrogate for Rank Loss. Because using Rank Loss or simple ranking loss in isolation significantly degrades performance and does not yield meaningful results, we report them in combination with ArcLoss/Cosine similarity for a fair comparison.
>
> | Ablation | PSNR | SSIM |
> | ----- | ------ | ------ |
> | Arc Loss     | 29.52   |  92.38     |
> | Cosine Similarity     | 29.13     | 91.10    |
> | Simple Ranking Loss  + Cosine similarity    | 28.32     | 89.92     |
> | Simple Ranking Loss  + Arc Loss       | 28.44     | 89.87     |
> | Rank Loss  + Arc Loss    | 29.83    | 92.74     |
>
>
> The comparison method uses the following formulas:
>
> **Cosine Similarity Loss:**
>
> $$
> 1 - \frac{\langle z_i, z_j \rangle}{|z_i||z_j|}
> $$
>
> **Simple Ranking Loss:**
> $$
> \mathcal{L}_{\text{pair}}(i,j)
> = \max\big(0, (d_j - d_i)\big)
> $$

---

> ### Author Response · Authors · 2025-11-21
> **Response to Reviewer TJiN Part 2**
>
> > 4. The method hinges on a learned AE latent; there is no analysis of how sensitive results are to the AE capacity or to using a
> stronger/pretrained encoder.
>
>
> We thank the reviewer for raising this point. We trained **three AE variants** with different depths and channel widths:
>
> * **3-level AE:** `[64, 128, 256]` (Our Choice)
> * **2-level AE:** `[64, 128, 128]`
> * **Shallow AE:** `[64, 128]`
>
>
> | AE Capacity      | PSNR | SSIM |
> | ---------------- | ---- | ---- |
> | `[64, 128, 256]` | 30.04  | 92.63  |
> | `[64, 128, 128]` | 29.89  | 92.55 |
> | `[64, 128]`      | 29.62  | 92.10  |
>
>
> We did not choose deeper or wider AEs because they led to GPU out-of-memory (OOM) on our A6000 (48 GB).
>
>
>
> We also evaluated two different **training volume sizes**, `64³` and `48³`.
>
> | Crop Size | PSNR | SSIM |
> | --------- | ---- | ---- |
> | `64³`     | 30.04  | 92.63  |
> | `48³`     | 29.51  | 91.71  |
>
>
> We chose `64³` as our default setting. Larger volumes, for example, `72³`, would also lead to GPU out-of-memory errors.
>
> We did not use a pretrained AE, since the AEs in other methods were also trained from scratch. However, using a pretrained AE (e.g., MAISI [1]) can bring notable improvements; for example, it increases the performance of the AE with channels `[64, 128, 256]` from 30.04 to 31.97.
>
> [1] Guo, Pengfei, et al. "Maisi: Medical ai for synthetic imaging." 2025 IEEE/CVF Winter Conference on Applications of Computer Vision (WACV). IEEE, 2025.
>
>
> > 5. The new metric is reasonable but mostly motivated empirically;
> there is no user/clinician study to show that better ∆-RMAE corresponds
> to more actionable longitudinal reads. If a disease progression does not have user study from medical doctors, it is not convincing to clinical
> readers.
>
>
>
> We  respectfully clarify that the $\Delta$-RMAE metric is **not purely empirical**: it is mathematically designed to *isolate longitudinal progression* by explicitly measuring the deviation of *changes over time*, rather than absolute values.
>
>
> This directly captures the clinically relevant notion of whether a method preserves *progression trends*, the primary focus of longitudinal analysis.
>
>
> Although we did not conduct a full-scale user/clinician study, the metric was **developed in collaboration with clinical members of the team**, who provided iterative feedback on its formulation. Several alternative metrics were prototyped, and $\Delta$-RMAE was selected specifically because it aligns with clinicians’ preference and validated in the experiments.
>
>
> We thank the reviewer for this constructive suggestion and plan to incorporate more extensive user studies with professional clinicians in future work.
>
>
>
> > 6. Clinical conditioning is mentioned (age, sex, status) but not
> deeply analyzed—no ablation on which attribute matters most, or on
> robustness to missing/noisy metadata.
>
>
> We thank the reviewer for the suggestion. We perform several analyses to validate robustness.
>
> **Age.**
> Age serves as the *temporal anchor* for longitudinal prediction (baseline age + time span). Injecting noise into age yields predictions that no longer correspond to a meaningful target timepoint, making such an ablation uninterpretable. For this reason, we do not treat age as a noise-robust attribute.
>
>
> **Sex.**
> We tested robustness by intentionally flipping the sex attribute. On ADNI, performance changed only slightly (PSNR/SSIM: 30.59/94.62 → 30.12/94.47).
>
>
>
> **Clinical status.**
> Clinical status is inherently noisy in real practice. To model this, we **explicitly added Gaussian noise during training** to reflect diagnostic uncertainty. The noise was chosen with mean 0 and standard deviation equal to one-third of the status class interval (interval = 3$\sigma$).
>
>
> Under perturbed status labels, performance remains reasonable (30.59/94.62 → 29.81/ 93.58), demonstrating robustness.
>
>
>
>
> Overall, although conditioning is not our main contribution, we demonstrate that the model is stable under realistic sources of metadata noise.  We will include these robustness experiments in the revised version.
>
>
>
>
>
>
> > 7. Many important previous works are not mentioned and compared, such as [1,2].
>
> Thanks for the suggestion. We are dicided to include the discussion of the paper mentioned [1,2], and also other papers [3,4]
>
>
> [1] Kyung, Daeun, et al. "Towards Predicting Temporal Changes in a
> Patient's Chest X-ray Images based on Electronic Health Records." arXiv
> preprint arXiv:2409.07012 (2024).
>
> [2] Liang, Kaizhao, et al. "Pie: Simulating disease progression via progressive image editing." (2023).
>
>
> [3] ImageFlowNet: Forecasting Multiscale Image-Level Trajectories of
> Disease Progression with Irregularly-Sampled Longitudinal Medical
> Images, ICASSP 2025 Oral.
>
> [4] Longitudinal Flow Matching for Trajectory Modeling, arXiv 2025.

---

> ### Author Response · Authors · 2025-11-26
> **Follow-Up on Our Previous Responses**
>
> We would like to once again express our sincere thanks for your time and thoughtful feedback.
>
> As the discussion deadline approaches, we would like to kindly confirm whether our responses have satisfactorily addressed your concerns, and whether there is anything further you would like to discuss.
>
> Best regards,
> The Authors

---

### Author Response · Authors · 2025-12-03
**Summary**

# Summary

We thank all reviewers for their time and constructive feedback, and the AC for their time and coordination.
Overall, the evaluations are mixed but generally positive. The paper’s scores increased from an initial 8-6-6-2 to 8-8-6-2. Due to the limited time, only one reviewer, CqTg, was able to engage with our response, provide further suggestions, and raise the score from 6 to 8.

We would like to express our gratitude to the reviewers for their appreciation of our work, specifically:

1. The novelty and coherence of the ArcRank loss and the reformulation of flow matching (dApu, UdDj), as well as their clear formulation (TJiN).
2. The elegant SVD-based alignment (UdDj).
3. The work is computationally interesting and scientifically important (CqTg).
4. Its clinical relevance, strong motivation, and that it is easy to follow (CqTg, UdDj).
5. The very good visual presentation (dApu, CqTg, TJiN), and the interpretability of the method (UdDj).


The reviewers provided many useful suggestions and feedback, mainly around:


1. **More ablations:** 1. SVD compared to the simpler angular/ranking formulation in ArcRank (TJiN, UdDj), and 2. isolated Arc/Rank loss with ([0, T]) ablation (dApu).

   **A:** We include this part of the ablation study in the revised version (Table 4).


2. **Runtime:** SVD runtime overhead and computational cost (TJiN, dApu).

   **A:** We have provided the results in Section 4.5 of the revised version. We use reduced SVD for computation, which gives a 6× speedup over full-matrix SVD. Overall, training with ArcRank loss takes about 40% more time than training without it.


3. **Stability:** SVD weights and stability (dApu).

     **A:** We provide, in the revised Section 4.4, a discussion of the selection and choice of weights and their influence on training stability.


4. **Uniform velocity assumption:** Temporal sampling [0, T] assumes uniform velocity (TJiN, UdDj). It would be beneficial to include more details on how conditioning on *T* and patient attributes relaxes this assumption (UdDj).


   **A:** This uniform limitation arises from the flow matching formulation, and fully addressing it is beyond the scope of this work. We recognize its importance and have discussed this limitation in the paper (as acknowledged by the reviewers). Furthermore, as suggested by UdDj, we provide an analysis of how conditioning on varying *T* results in different estimated velocity fields, which partially relaxes the uniform-velocity assumption.


4. **Comparative study:** More comparative study with ImageFlowNet (CqTg).

   **A:** We include the results in Tables 2 and 3.


5. **AutoEncoder:** Influence of AE capacity (TJiN).

   **A:** We provide an ablation on the influence of AE capacity in Table 5 of the revised paper.


6. **Metric:** $\Delta$-RMAE reasonableness and robustness (TJiN, CqTg), and sensitivity to misregistration (UdDj).

   **A:** We provide a discussion on metric stability in Section E and include Table 7 to analyze its sensitivity to misalignment. The $\Delta$-RMAE metric is developed mathematically and in collaboration with clinical team members, who provided feedback on its utility for modeling disease progression.


7. **Clinical condition robustness (TJiN).**

   **A:** We have included the results in Section F.

8. **Dataset limitations.** Evaluation is confined to AD-style neurodegeneration datasets (TJiN).

    **A:**  Auto-Encoder and Flow-Matching models require a large amount of data for stable training. The three AD datasets provide a large cohort (around 2,000 volumes) that supports this training. We are also currently preparing osteoarthritis (OA) and glioblastoma (GBM) cohorts from multiple sources and are doing data standardization and registration, which is very laborious. The challenge is that registration for OA is difficult because X-rays (used for OA diagnosis) are not acquired on the same plane. For GBM, the progression analysis dataset uses diverse MRI acquisition protocols, which are hard to standardize. We are still preparing these datasets, and including results on them is our next planned step.


---

We hope that the summary will assist the AC in making the final recommendation. We appreciate the significant efforts that the AC and PC have made during this challenging period.

Best regards,

The Authors

---

### Meta-Review · Area_Chair_VS1r · 2026-01-09

**Summary:**

The paper proposes a latent flow-matching framework for patient-specific disease progression modeling from longitudinal image data. By treating disease progression as a continuous velocity field in latent space, the method is well motivated clinically and obtains patient-specific latent alignments. The key contribution is introducing the ArcRank loss to encourage consistent trajectory direction and increasing severity, aligning latent dynamics with individualized temporal modeling and yielding interpretable latent flows for disease progression. The experiments on multiple datasets demonstrate good performance.

Reviewers raise questions and concerns about ablation studies with auto-encoders, baseline comparison, dataset limitations, computational cost, and assumptions. The author provides a thorough response with detailed clarification and new results, which answer the questions well.

After rebuttal, one out of the four reviewers responded and raised the score from 6 to 8. Other reviewers weren’t able to participate in the discussion unfortunately.

Overall, I think the paper studies an important problem motivated by clinical applications, and proposes a well-motivated method that may bring interesting insights to the community bringing methodology and applications. I would recommend accept.

**Reviewer Concerns:**

Reviewers raise questions and concerns about ablation studies with auto-encoders, baseline comparison, dataset limitations, computational cost, and assumptions. The author provides a thorough response with detailed clarification and new results, which answer the questions well.

**Reviewer Scores:**

After rebuttal, one out of the four reviewers responded and raised the score from 6 to 8. Other reviewers weren’t able to participate in the discussion unfortunately.

---

### Decision · Program_Chairs · 2026-01-26

Accept (Poster)